Subject Areas:
biomechanics

Keywords:
locomotion, rotatory treadmill, substrates

Author for correspondence:
David C. Lin
e-mail: davidlin@wsu.edu

# Comparison between the kinematics for kangaroo rat hopping on a solid versus sand surface

Joseph K. Hall[1], Craig P. McGowan[2,4,5,†] and David C. Lin[1,2,3]

[1]Voiland School of Chemical Engineering and Bioengineering, [2]Washington Center for Muscle Biology, and [3]Department of Integrative Physiology and Neuroscience, Washington State University, Pullman, WA, USA
[4]School of Biological Sciences, University of Idaho, Moscow, ID, USA
[5]WWAMI Medical Educational Program, Moscow, ID, USA

 DCL, 0000-0003-4492-0944

In their natural habitats, animals move on a variety of substrates, ranging from solid surfaces to those that yield and flow (e.g. sand). These substrates impose different mechanical demands on the musculoskeletal system and may therefore elicit different locomotion patterns. The goal of this study is to compare bipedal hopping by desert kangaroo rats (*Dipodomys deserti*) on a solid versus granular substrate under speed-controlled conditions. To accomplish this goal, we developed a rotary treadmill, which is able to have different substrates or uneven surfaces. We video recorded six kangaroo rats hopping on a solid surface versus sand at the same speed ($1.8 \, \mathrm{m \, s^{-1}}$) and quantified the differences in the hopping kinematics between the two substrates. We found no significant differences in the hop period, hop length or duty cycle, showing that the gross kinematics on the two substrates were similar. This similarity was surprising given that sand is a substrate that absorbs mechanical energy. Measurements of the penetration resistance of the sand showed that the combination of the sand properties, toe-print area and kangaroo rat weight was probably the reason for the similarity.

## 1. Introduction

Animals locomote on a variety of terrains in their habitats, moving on surfaces that may have substantially different mechanical properties. A solid substrate, such as hard-packed dirt, is a relatively rigid surface that animals lose minimal energy to during impact or during push-off, while granular substrates,

[†]Present address: Department of Integrative Anatomical Sciences, Keck School of Medicine of the University of Southern California, Los Angeles, CA.

such as sand, are flowable media which absorb energy during penetration. For example, walking on sand requires 1.6 to 2.5 times more total work to be produced for humans versus walking on concrete [1]. In running zebra-tailed lizards, it was estimated that the hindlimb muscles performed three times more mechanical work on a granular substrate than on a solid surface [2]. Therefore, different substrates impose different mechanical demands on the musculoskeletal system, and animals may adapt their motor strategies to meet the changes in terrain substrate.

Evidence for changes in motor strategy due to substrate properties can be provided by kinematics studies comparing locomotion on different substrates. For gymnophthalmid lizards running on walkways, differences in stride length and frequency have been observed for matched speeds on sand and solid substrates [3]. However, this study examined burst locomotion, implying that the movements may not have been during steady locomotion, thus influencing their kinematic patterns. In a study of zebra-tailed lizard running, stride length decreased by 15% on the granular versus solid substrate, and the limb posture shifted from a digitigrade to a plantigrade position on solid versus granular substrate, respectively [2,4]. In ostriches, the kinematics of the toe joints were different while running on sand versus solid substrates, with a greater range of motion on the sand surface [5]. In a study using a hopping robot [6], the optimal strategy to achieve the highest height in a landing-jumping task was found to include a time delay between landing and push-off to avoid the transients associated with the relaxation of the granular media into a compacted state. Because of the similarity of this robotic task to the impact of landing and push-off during animal hopping, a similar strategy could be employed, changing the duty cycle defined by the ground and aerial phases of the kinematics. However, hopping on different substrates has not been well studied in animals.

Kangaroo rats are bipedal hoppers that encounter both solid and granular terrains in their desert habitat. The goal of this study is to compare bipedal hopping by kangaroo rats on solid versus granular substrates under speed-controlled conditions on a custom-designed treadmill. Specifically, we tested whether the different substrates changed the kinematics of hopping, as suggested by the lizard and robot studies. However, two alternatives were possible. The animals could maintain similar general kinematics of locomotion (i.e. hop period, hop length and duty cycle) by compensating for changes in the mechanical demand of different substrates, or the combination of contact surface area (i.e. toe-print area), sand properties and animal weight was such that animals did not need to compensate for more energy dissipation from increased penetration into the granular substrate [7]. The significance of this study is that it represents the first of its kind to determine the effects of substrate on the kinematic patterns of bipedal hopping, which will provide insight into the adaptations that animals make while locomoting on different substrates.

# 2. Material and methods

## 2.1. Animals and experimental protocol

Six desert kangaroo rats (*Dipodomys deserti*) were wild caught in the Mojave Desert in southeast Nevada (mass: 91–121 g) in accordance with permits issued by the Nevada Department of Wildlife (License 466453) and the Bureau of Land Management. Animal husbandry and all animal experimental procedures were approved by the University of Idaho Institutional Animal Care and Use Committee (#2016-68). The joint centres of the left hindlimb (hip, knee, ankle and metatarsophalangeal (MTP)) were marked with a white and black spot with the combination of 'white out' correction fluid and black marker, and the limb segments measured.

To study the effect of different substrates at controlled speeds, we developed a rotary treadmill which we call the variable terrain rotary treadmill (VTRT). Briefly, the VTRT consisted of a large wooden disc (radius = 1 m) that was supported by wheeled casters (figure 1a). A motor (Part#X3573-MTE3847-1; Motion Control Group) with velocity servo (model B30A8; Advanced Motion Controls) was coupled to the rim of the disc via a small inflatable tyre attached to the shaft of the motor. The solid surface was the wooden disc covered by latex paint and the granular surface was sand obtained from the animals' desert habitat. The sand depth was 2.5 cm, which was sufficient to prevent 'bottoming out', confirmed by inspection of footprints, and we later verified that this depth was sufficient to avoid interaction with the solid surface below (see Characterization of sand properties, below). A plexiglass enclosure was held slightly above the surface of the VTRT (figure 1a) so the animal was positioned in front of the video camera (focal length 2 m) with a capture rate of 200 Hz and a resolution of 640 × 480 pixels (XC1-M; Xcitex). The camera was positioned 2 m from the enclosure to minimize the effects

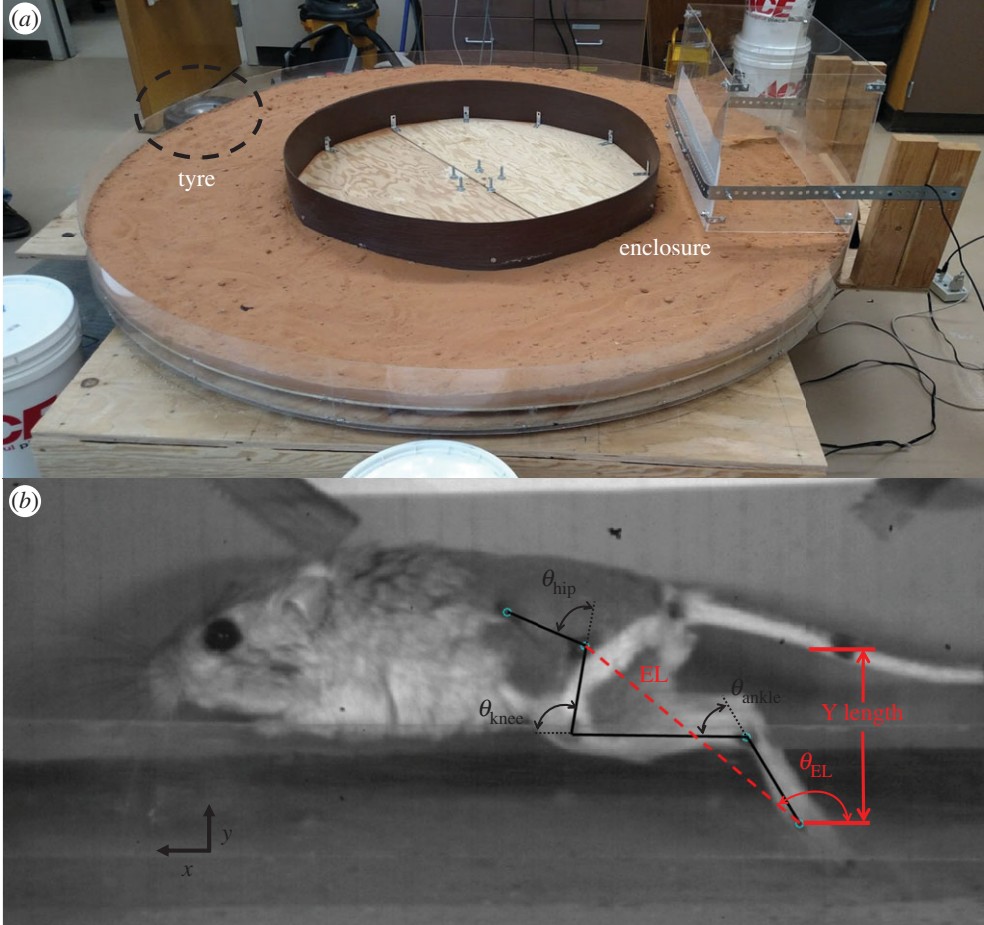

**Figure 1.** (*a*) View of VTRT with enclosure and tyre coupled to motor servo. (*b*) Kangaroo rat on VTRT with effective limb (EL) (dashed red line) vector shown. The EL angle ($\theta_{EL}$) was measured with respect to the horizontal, and the Y length was measured as the vertical distance from the MTP and hip joints. Also shown are the hip, knee and ankle angles ($\theta_{hip}$, $\theta_{knee}$ and $\theta_{ankle}$, respectively), defined as external joint angles.

of parallax and calibrated for spatial measurements (pixels to mm). A brush attached to the front of the enclosure ensured that the animals were hopping on an even sand surface.

Trials began with the VTRT at rest, and data collection began when the animal achieved steady bipedal hopping. Animals performed two or three trials at each speed with a target of recording 10 steady hops. Because we did not know *a priori* the animal's preferred hopping speed (at which hopping was most consistent; see below for criteria for valid hops), data were collected for five speeds ranging from 1.25 to 2.25 m s$^{-1}$ in 0.25 m s$^{-1}$ increments.

## 2.2. Kinematic data analysis

Videos were first visually inspected for valid hops. The criteria were hopping on the hindlimbs only; simultaneous bilateral landing and taking off; a sequence of three consecutive good hops and not moving forwards or backwards relative to the camera. Individual hops were defined by the time the animal first touched the surface to the next time it touched down. Take-off was defined as the time the toes left the surface.

Markers for the pelvis, hip, knee, ankle and MTP joints were digitized using Matlab code [8] (figure 1*b*). The toe marker was not digitized because the toe was obscured by the sand while the animal was on the surface. Joint angles were then calculated from the digitized markers and the limb lengths of the individual animal. Due to the movement of the skin over the knee, the knee joint angle was estimated from the positions of the hip and ankle markers and the lengths of the femur and tibia using the law of cosines [9,10]. Further, any error in the estimate of knee angle would not accumulate over time due to the cyclic nature of the movements. Skin markers over the ankle and hip joints had

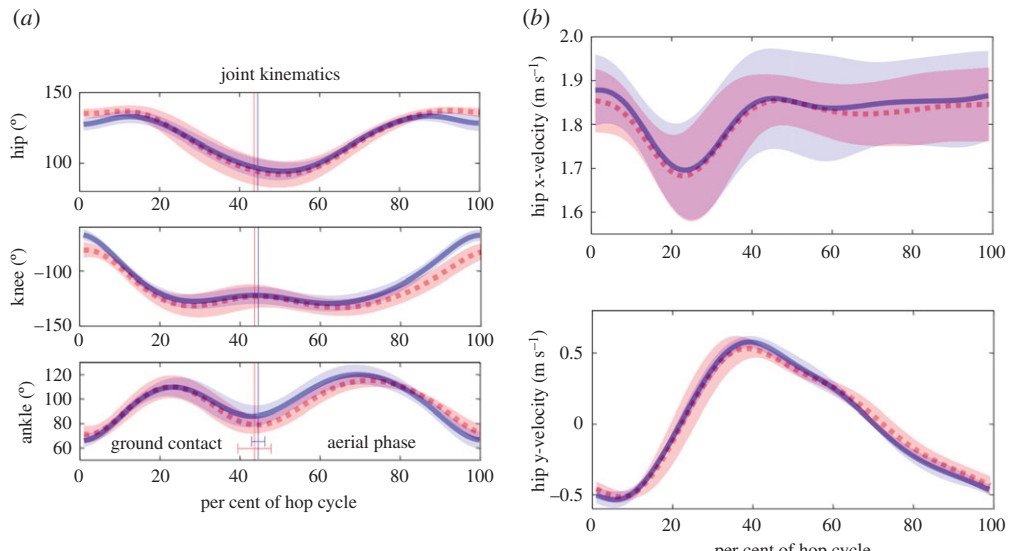

**Figure 2.** (a) Average hip, knee and ankle angles versus hop cycle on solids (solid lines) and sand (dashed lines) surfaces, with shading representing ± one standard deviation ($n = 6$ animals; 43 total trials for solid surface and 40 trials total for sand surface). The average take-off times for the solid and sand surfaces are shown with the error bars representing ± one standard deviation. (b) Hip x-velocity and y-velocity versus hop cycle ($n = 6$ animals; 43 total trials for solid surface and 40 trials total for sand surface).

minimal random motion artefact because the skin is relatively tight over those joints, thus any error in the knee angle estimate should not accumulate spatially. The joint angles were filtered with a fourth-order low-pass Butterworth filter (cut-off of 25 Hz which was sufficient for the 6 Hz hopping frequency (see Results)). To compare kinematic data between animals and hops, we normalized each hop period to 100% by interpolating data between frames, resulting with each hop represented by 101 points (figure 2).

Due to the approximately 100 kg additional inertia of the sand, the dynamics of the VTRT to reach steady-state speed were slower than without sand, and some hops were recorded before steady-state speed was achieved (about 12% for the sand trials only). To ensure that hop velocities were matched between the hard and sand surfaces, we first calculated the speed of the MTP marker during touchdown and assumed that the foot was not moving with respect to the VTRT during this time interval (i.e. the foot speed was equal to the VTRT speed). This assumption presumes that there was minimal slippage of the foot on both surfaces, but this could not be verified on sand due to the sand obscuring visualization of the entire foot. This method was used because we could not see identifiable markings on the VTRT on every trial. In trials where identifiable markings on the VTRT were observable, the estimate of VTRT speed from the foot agreed with estimate from the VTRT markings. Thus, to estimate the speed of the animal, we added the speed of the hip marker (averaged over the entire hop cycle) to the foot speed. The average hip speed was used as a surrogate for the speed of the centre of mass (COM) of the animal (see Results). In compiling all of the trials which classified as valid trials, we determined that $1.8 \pm 0.3 \ \mathrm{m \ s^{-1}}$ (±16%) was the speed which produced the most hops deemed as valid and was used for speed-matched comparisons.

For the trials matched by our estimate of the speed of the animal (see previous paragraph), we calculated several metrics of hopping kinematics and averaged their values to estimate the overall hopping metrics for each animal. These metrics were hop period, hop length and duty cycle (time on the surface divided by hop period). We also calculated the effective limb (EL) vector (MTP marker to hip marker) (figure 1b). Because of the relatively small sample size ($n = 6$), we used the non-parametric two-tailed Wilcoxon signed-rank test (comparable to the parametric paired $t$-test) to test for differences between hopping on solid and sand substrates. Results were considered significant at $p < 0.05$.

## 2.3. Characterization of sand properties

To characterize the properties of the sand used in the experiments, we first measured the angle of stability by placing and levelling sand horizontally on an aluminium tray and tilting the tray until the sand began to flow [11,12]. The angle of stability was found to be 33°. Next, we estimated the bulk density of the sand by weighing a 50 ml graduated cylinder filled with sand [12]. The bulk density was calculated to be

1.56 gm cm$^{-3}$. Note that angle of stability and bulk density differ from the properties of angle of repose (the steepest angle at which the material can be piled) and material density (bulk density divided by volume fraction), respectively.

To verify that the depth of sand on the treadmill was sufficient to prevent boundary effects of the solid surface below, we made sand penetration measurements for different sand depths, similar to the testing protocol of Li *et al.* [2]. Specifically, a custom-made penetrator was mounted on the arm of a servo system with integrated force transducer (model 309C; Aurora Scientific; see technical note 300C-I-T01 for details on the set-up). The penetrator was made from wood and had the rectangular dimensions of 2.5 × 1.25 cm, which was based on dimensions of an ink print of the toe segment of an average-sized kangaroo rat. The servo arm and penetrator were rotated by the servo motor to create a near linear motion because of the small amount of rotation needed (less than 1°). During steady speed hopping, only the toes contact the surface. To replicate the landing of the toes on the sand during hopping, we set the servo with the servo software input command to a constant velocity of 50 cm s$^{-1}$ for the penetrator, which was the average vertical velocity of the foot at landing. The penetration force was limited to 3.0 N by a dial setting on the servo, which was based on the maximum ground reaction force data per leg observed during kangaroo rat hopping [13]. Penetration tests were repeated five times for sand depths of 0, 0.5, 1.5, 2.0, 2.5, 3.0, 3.5, 4.0 and 5.0 cm. To increase the consistency of the measurements, the penetrator was positioned above the sand surface and slowly lowered until a very small pre-load (0.2 N) was reached. This procedure and amount of pre-load made the surface more uniformly flat because the small pre-load smoothed out any irregularities while not compressing the sand. The sand was aerated after each trial by gentle raking and then re-levelled.

To quantitatively characterize the sand, a measurement of the penetration resistance (PR) was made for each trial of data. Because the plot of force versus length was roughly linear until the force was 2.5 N in all trials (see Results), PR was estimated as

$$PR = \frac{(2.5 - F(t = 0))}{(X(F = 2.5) - X(t = 0))},$$ (2.1)

where PR is the PR (N mm$^{-1}$), $F$ is the measured force (N), $X$ is the position of the penetrator (mm) and $t$ is time. Note that the numerator of equation (2.1) is the difference of 2.5 N and the initial force (about 0.2 N) and the denominator is the difference between the position when the $F = 2.5$ N and the initial position of the servo. We determined whether there was a significant difference in the PR for the sand depths by using an ANOVA followed by multiple pairwise comparisons (Tukey's honest significant difference criterion; 'multcompare' in the Matlab statistics toolbox (Matlab 2019b; Mathworks)). Significance level was taken to be 0.05.

## 3. Results

For the hopping speed of 1.8 m s$^{-1}$, which was close to the preferred speed for desert kangaroo rats in the wild [14], the number of trials which met the criteria for inclusion ranged from four to nine hops for individual animals. The hop periods were not significantly different (averages ± standard deviations for sand versus solid substrates were 0.162 ± 0.012 and 0.155 ± 0.001 s, respectively; $p = 0.22$). Further, there were no significant differences between the hop lengths (sand versus solid, 0.293 ± 0.024 versus 0.281 ± 0.017 m, respectively; $p = 0.69$) and duty cycles (sand versus solid, 0.435 ± 0.042 versus 0446 ± 0.016, respectively; $p = 0.69$). Thus, the general kinematic patterns of hopping on sand and solid substrates were similar.

During hopping, the horizontal velocity of the COM should decelerate upon landing, accelerate toward take-off, and be near constant in the aerial phase [9]. We found that the hip marker was the closest to this pattern (figure 2). Therefore, we used the hip kinematics as an approximation of the COM. Although the actual COM lies approximately 2–3 cm anterior to the hip in animals with a crouched posture, the COM position probably does not shift substantially relative to the hip position during hopping, thus we assumed that the velocities and accelerations should be similar for the hip and COM. Both the horizontal and vertical velocities of the hip for sand and solid substrates were comparable (figure 2). For the ground contact phase of the hop, we found no significant differences for minimum horizontal velocity, maximum horizontal deceleration and acceleration, and minimum and maximum vertical velocities of the hip (table 1). Thus, the overall kinematics of the COM was also similar for hopping on sand versus solid substrates.

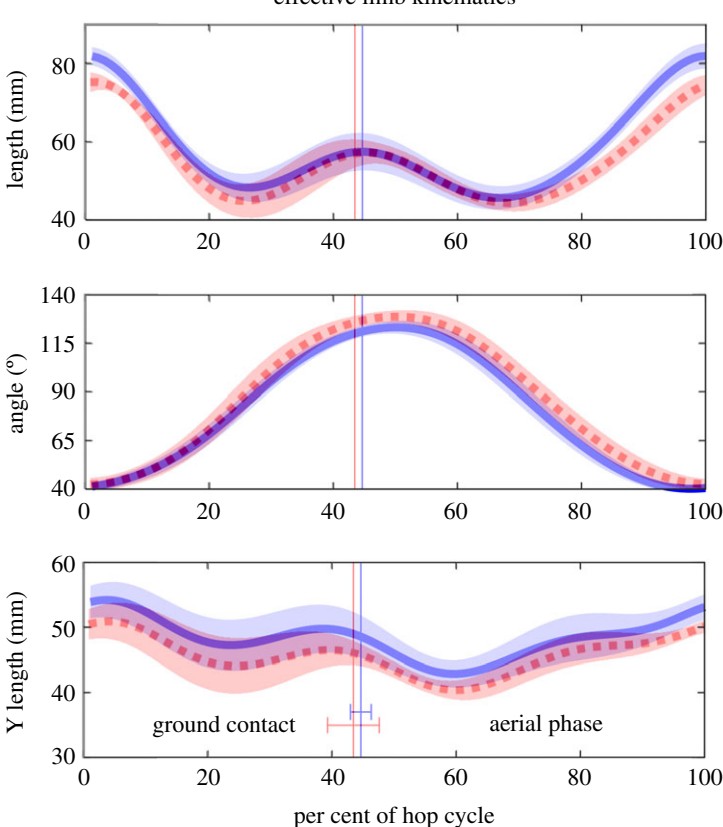

**Figure 3.** Comparison of EL kinematics between hopping on sand and solid substrates. See figure 1b for definition of variables plotted.

**Table 1.** Comparison of kinematic variables of the hip between the sand and solid substrate conditions.

|  | sand | solid | *p*-value |
|---|---|---|---|
| minimum horizontal velocity (m s$^{-1}$) | 1.65 ± 0.09 | 1.66 ± 0.10 | 1.0 |
| minimum horizontal acceleration (m s$^{-2}$) | −10.56 ± 1.93 | −11.78 ± 3.04 | 0.44 |
| maximum horizontal acceleration (m s$^{-2}$) | 10.65 ± 1.48 | 9.87 ± 0.97 | 0.16 |
| minimum vertical velocity (m s$^{-1}$) | −0.54 ± 0.05 | −0.57 ± 0.06 | 0.31 |
| maximum vertical velocity (m s$^{-1}$) | 0.58 ± 0.09 | 0.60 ± 0.04 | 0.56 |

Although there were no significant differences in the gross kinematics metrics that we measured, there were some differences that were visually apparent. Namely, the joint kinematics of hopping on sand appeared to be more flexed. To quantify this observation, we calculated the EL vector (figures 1b and 3). We found a significant difference for the EL length at landing (sand and solid, 75.1 ± 2.5 and 82.0 ± 2.3 mm, respectively; $p = 0.0313$) which was due to more flexed hip, knee and ankle angles on sand ($p = 0.0313$ for all measures) (figure 2a). The difference in EL length at mid-stance was more ambiguous and did not reach the level of statistical significance, with a value of $p = 0.0625$ (sand and solid, 46.1 ± 3.8 and 49.6 ± 3.4 mm, respectively). At take-off, the EL lengths were not significantly different, but the EL angles were significantly different ($p = 0.6875$ and 0.0313, respectively). This more protracted limb angle coincided with a noticeable cloud of sand being kicked up after take-off, indicating that work was being done on the sand. Lastly, the more crouched posture during hopping on sand was shown by a significant difference in the average vertical EL lengths during the entire hop ($p = 0.0313$) (figure 3).

The sand penetration experimental set-up measured force and position of the penetrator over time, and force was plotted versus position from which the PR was estimated (figure 4a). The PR was highest for the wooden surface (0.0 mm of sand) and decreased as the sand depth increased until a

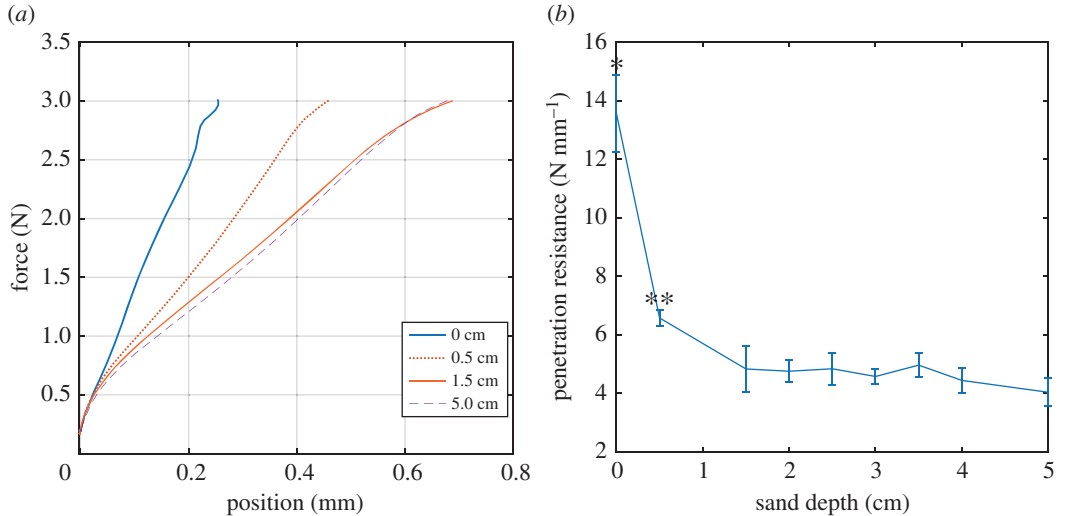

**Figure 4.** Experimental measurement of the PR of the sand used in the study. (a) Representative plots of force and penetrator position for four different sand depths. (b) Measurement of PR for sand depths of 0, 0.5, 1.5, 2.0, 2.5, 3.0, 3.5, 4.0 and 5.0 cm. The PR for the sand depths of 0.0 and 0.5 cm was significantly different to the PRs of the other depths.

depth of 1.5 mm. Increasing the depth beyond 1.5 mm did not change the PR (figure 4b). The ratio of the PR of the wood surface to PR of the sand depth used in the experiments (2.5 cm) was 2.8 : 1. The ANOVA with sand depth as the factor was statistically significant ($p < 0.00001$) and the pairwise comparisons showed that PRs for sand depths of 0.0 and 0.5 were statistically different to the PR for any other depth, and the PRs for any sand depths greater than or equal to 1.5 mm were not statistically different. These results support that the sand depth of 2.5 cm used was sufficient to avoid any boundary effects of the wooden surface below the sand.

## 4. Discussion

Hopping on sand is a difficult challenge for the musculoskeletal system. Although we do not have direct measurements of the energy lost to the sand, which would require tracking the sand motion and force plate data [1], we expected that hopping on sand imposed more mechanical demands on the animals for two reasons. First, a previous study of zebra-tailed lizards estimated that muscles do three times the amount of work while the animals run on a fluidized bed of granular material versus on a solid surface. Second, the PR of the sand was almost three times less than that of the solid substrate. Moreover, the sand used in these experiments had a stability angle similar to that of the most 'unstable' granular substrate tested (34°) for a wide range of particle sizes [12], implying that the sand was easily displaced. Although bipedal hopping differs substantially for other modes of locomotion studied, these two reasons suggest that the animal would be doing work on the sand and as a result, we predicted that the kinematics of the locomotion pattern would change with the change in substrate, similar to other animals [2,4,7,12].

To test this prediction, we developed the VTRT to enable speed-controlled experiments on different terrains (substrates or slopes) which are not possible on traditional belted treadmills. We only used two different substrates for this study, but rocky terrains and slopes can also be implemented. However, the VTRT had two limitations. First, not unexpectedly, the settling time of the velocity servo during the initial start-up was considerably longer with the addition of the large inertia of the sand. Thus, we chose to estimate the speed of the animal from the recorded kinematics of the hip and MTP markers. Second, due to rotation of the disc, animals experienced centrifugal force. From the mass of the animal, distance of the animal from the axis of rotation and angular velocity, we estimated the centrifugal force to be approximately one-third of their body weight at the speed tested. While this amount of force could influence the movement, the animals did not noticeably reorient their bodies (i.e. they did not hop toward the centre) or change their trajectories (i.e. they did hop toward the circumference). This assumption would have to be confirmed by three-dimensional video recordings.

Our study showed that the overall pattern of hopping, as described by the hopping metrics and COM kinematics, was similar between solid and sand surfaces, which contradicted our prediction about the

effect of a sand substrate on hopping kinematics. There are two potential mechanisms for the lack of change in kinematics. One potential mechanism is that some neuromuscular compensation occurred to offset the additional mechanical demands of the sand substrate. There is some indication that neuromuscular compensation did occur in that the limb posture, as assessed by the EL length, and was found to be different at landing and during mid-stance (although not quite reaching statistical significance for mid-stance). This result indicates that animals assumed a more crouched posture on sand versus the solid substrate. These results are consistent with a study of human walking on uneven terrains, which showed no change in gait metrics but a more crouched posture [15,16]. In an insect study, decreases in the vertical position of the body COM occurred while running on more compliant surfaces, similar to the more crouched posture observed in this study [17]. The hypothesis that the more crouched posture while hopping on sand is an active neuromuscular adaptation needs to be tested with additional measurements, such as electromyography, to determine the amount of neural compensation.

The second potential mechanism for the lack of changes in hopping kinematics is that neuromuscular adaptation was not needed due to the physical characteristics of the sand, the area of toe print during the contact phase and weight of the kangaroo rats. With regard to the sand properties, variations in sand composition, granule size and granule shape all influence different mechanical characteristics [18]. Our measurements of the sand from the kangaroo rats' habitat included angle of stability, bulk density and PR. Interestingly, in a survey of different sand with a specific range of particle sizes, the angle of stability (33°) and bulk density (1.56 gm cm$^{-3}$) of the sand used in our study were very similar to the sand (34° and 1.532 gm cm$^{-3}$, respectively) on which steppe-runner lizards had the highest locomotor performance (maximum velocity and acceleration) [12]. In addition, the landing phase of hopping on sand may be similar to the 'slap phase' of locomotion on flowable surfaces, including sand and water, in which the high-speed downward impact solidifies the media [6,19]. Thus, it is possible that the properties of the Mojave sand, in combination with hopping kinematics, are well suited as a substrate for high-performance locomotion to occur on.

A detailed study of the interdependence of sand properties, contact area of the foot, and animal weight and limb length for locomotor performance was conducted by Qian *et al.* [7] using a fluidized bed of granular material. This study used a hexapod robot, three species of lizard, geckos and ghost crabs to measure the locomotor performance for a diverse range of anatomical morphologies and sizes on a wide range of substrates, with the objective of generalizing how substrate properties, morphology and size interact to affect locomotor performance. A unitless parameter, called the leg penetration ratio, $\tilde{d}$, was formulated as

$$\tilde{d} = \frac{F}{(k * l)},$$ (4.1)

where $F$ is the force exerted on contact area, $k$ is the PR and $l$ is the length of the limb. For applying equation (4.1) in the Qian *et al.* study, $F$ was taken as the weight of the animal, $k$ is an experimentally based parameter that depends upon the substrate and scales linearly with contact area [4], and $l$ was taken as hip to ankle length. The importance of $\tilde{d}$ is that for $\tilde{d} < 1$ (called the 'insensitive region'), average locomotion speed was unaffected by the substrate. Conversely, for $\tilde{d} > 1$, increasing values of $\tilde{d}$ resulted in a steep decrease in average locomotion speed.

For kangaroo rats hopping on Mojave sand, we used a value of $F = 3.0$ N because that was the maximal ground reaction force observed [13], $k = 4.8$ N mm$^{-1}$ was the value found for the sand depth of 2.5 cm (figure 4$b$), and $l = 50$ mm was approximately the minimum EL length recorded (figure 3), resulting in a value of $\tilde{d} = 0.013$. Note that although we used values of $F$ and $l$ that maximized the value of $\tilde{d}$, the value of $\tilde{d}$ was still substantially less than the critical value of 1. We also note that the value of 4.8 N mm$^{-1}$ for PR measured in this study is substantially greater than the measured values for sand (29 Palms sand) in a previous study [4], which was approximately 2.6 N cm$^{-1}$ when scaled by the area of our penetrator. Major differences in the testing set-up between the previous study and our study could be responsible for this discrepancy, namely the speed of the penetrator (50 cm s$^{-1}$ (this study) versus 1 cm s$^{-1}$) and the sand being fluidized in the previous study. However, even with using the substantially smaller value of 2.6 N cm$^{-1}$ for PR, the value of $\tilde{d}$ is equal to 0.23. Lastly, we note this analysis is limited by potential differences between our testing method and the natural toe–sand interaction. First, the penetrator was made of solid wood, which will interact with the sand differently to the more flexible kangaroo rat toes. Second, rotation of the toes, if it occurs during hopping, may produce a different penetration effect to the perpendicular intrusion of the penetrator

used. More detailed analyses of *in vivo* foot–substrate interactions are needed to address these uncertainties.

The calculated value of $\tilde{d}$ provides strong support that the properties of the Mojave sand, toe-print contact area and weight of the animal (as reflected in the value of $F$) combine to make kangaroo rat hopping performance insensitive to the sand substrate. Specifically, the size of toe-print area plays an important role because larger area increases the PR proportionally. In desert kangaroo rats, the toe segment dimensions were measured to be $2.5 \times 1.25$ cm, approximately 16% of their total leg length (summed segment lengths) [20], and all kangaroo rat species have a greatly elongated foot compared with similarly sized quadrupedal mammals [14]. Potentially important is that the toes of desert kangaroo rats are covered in dense fur, filling the space between the toes and increasing the total surface area. More generally, this species is highly specialized because it is only found in habitat with soft sand dunes [21]. This study confirms that desert kangaroo rats have morphological adaptations that enable them to manoeuvre through soft sand with high locomotor performance.

Ethics. All experimental procedures were approved by the University of Idaho Institutional Animal Care and Use Committee.

Data accessibility. Data collected are available from the Dryad Digital Repository: https://doi.org/10.5061/dryad.mgqnk990x [22].

Authors' contributions. All three authors carried out the experimental design and protocol, data analysis, and drafting the manuscript. All authors gave final approval for publication.

Competing interests. We declare that we have no competing interests.

Funding. Funding from the Army Research Office (grant no. 66554-EG to D.C.L. and C.P.M.), National Science Foundation (grant no. 1553550 to C.P.M.) and Auvil Scholarship (to J.K.H.).

Acknowledgements. The authors would like to acknowledge the assistance of Drs Clint Collins, Jeffrey Rankin, Marie Schwaner and May Qiang.

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
