## [Peer Review File · Royal Society Open Science]

Review History

RSOS-201796.R0 (Original submission)

Review form: Reviewer 1

Is the manuscript scientifically sound in its present form?

Yes

Are the interpretations and conclusions justified by the results?

Yes

Is the language acceptable?

Yes

Do you have any ethical concerns with this paper?

No

Have you any concerns about statistical analyses in this paper?

No

Recommendation?

Accept with minor revision (please list in comments)

Comments to the Author(s)

The authors have generally responded well to the reviews received when their MS was submitted to Biology Letters and have clarified several matters that were not clearly presented when first submitted. They have also provided a reasonable argument in response to the reviews for the limited kinematics analysis that they perform, in the absence of ground force recordings (which would be quite difficult with their setup), EMGs of muscles and any in vivo muscle forces. They show that basic very general kinematic patterns of the hindlimbs are not significantly different hopping on sand vs a solid substrate. I do think, however, that they need to quantify EL length at mid-support for both condition in the context of arguing that the limb appears more flexed. This may be the case at landing, but the joint angle patterns shown in Fig. 1C do not support this being the case over much of the support phase. Improved explanation and discussion of this is the only main issue I have regarding the revised MS and the changes made by the authors regarding their findings.

Specific comments:

l. 56-58 "In this previous study on lizards, the locomotion was performed in bursts, implying that the movements may not have been during steady locomotion, thus influencing the kinematic patterns."

Suggest rephrase as "However, locomotion was performed in burst in this previous study of lizards, implying that the movements may not have been during steady locomotion, thus influencing their kinematic patterns."

l. 61-61 "Because of the similarity of the task in this previous study to the impact of landing and pushoff during animal hopping"

Suggest rephrase as: "Because of the similarity of this robotic task to the impact of landing and pushoff during animal hopping"

l. 63-64: last new sentence - this would seem better made as the topic sentence of the paragraph not as its conclusion.

l. 70 suggest: "maintain similar general"

l. 84 "metatarsophalangeal"

l.99-102 "and calibrated for spatial measurements" repeated twice in these sentences

l.160 "patterns" pl. not singular

l.166 horizontal and vertical velocities of ?? (hip as proxy for COM, I believe)

l.168: ditto - presumably these are for hip as COM estimate

l.173 "gross kinematics" add "that we measured"

l.174-177: the authors state that the limb appears more flexed hopping on sand, but the joint angles shown in Fig. 1C don't clearly demonstrate this. To quantify this, they report EL length "at landing". However, this only reflects how flexed the joints may be at landing; not over the course of stance or mid-support when the greatest GRF would be expected to act. The authors

should also test for whether EL length differs at mid support to better evaluate the the limb is more flexed in relation to greater joint loading (rather than just a landing and takeoff). Clearer explanation and rationale for these statements and what is shown in Fig. 1C is therefore needed.

l.188 rephrase as: “we presume that hopping on sand imposed”

l.194-195 suggest rephrase as “Thus, it is likely that some neuromuscular compensation occurred to offset...”

l.196 - break into 2 sentences. “substrate. This hypothesis...”
Also “in vivo force transducers” force of what?? (VAGUE)

l.208-210 re: centrifugal force being 1/3 body weight - I would suggest this is a good place to acknowledge limitations of 2D kinematics analysis and that 3D kinematics would help to address any ML compensation by the animals.

Figures

Sample sizes should be given for the plots in 1C and Fig. 2A,B&C. In Figs 1C & 2C, it would help to identify the stance vs swing phases, even though ‘takeoff’ is labeled. Also, I don’t find the light orange - brownish trace for sand very clear - if made darker or a brighter color it would show up better.

Review form: Reviewer 2

Is the manuscript scientifically sound in its present form?

No

Are the interpretations and conclusions justified by the results?

No

Is the language acceptable?

No

Do you have any ethical concerns with this paper?

No

Have you any concerns about statistical analyses in this paper?

No

Recommendation?

Reject

Comments to the Author(s)

This study uses a rotary treadmill and high speed imaging to record and compare the kinematics of kangaroo rats hopping on sand vs. solid surfaces. The authors found that the gross kinematic variables are similar, but a few leg kinematic features are different.

The study is novel in that it is the first to compare bipedal hopping kinematics on sand vs. solid surfaces. However, it suffers several major issues that must be addressed before further consideration.

1. It is not clear if the very shallow sand bed tested really behaves like sand or has substantial boundary effect from the bottom. The kangaroo rat's leg length is on the order of 10 cm (from Fig. 2C), and its foot alone is probably 3 cm long. By contrast, the sand bed is merely 2.5 cm deep. It will not be surprising if the toes can reach the rigid bottom, if the foot undergoes substantial rotation subsurface (see Li, Hsieh, Goldman, 2012, JEB, Fig. 11C). Even if it does not, there can be substantial forces from the solid boundary. This is a well-known effect in confined granular media. Even when an intruding object is still at a substantial distance from the walls that contain granular media, forces already start to go up substantially (e.g., see Li, Zhang, Goldman, 2013, Science, Fig. 2B and Materials & Methods first section). My guess would be that the near solid bottom effect due to the very shallow depth of the sand largely contributes to the similar overall kinematics observed. Why not try it again with deeper sand (~10 cm or above)? It looks like the treadmill can well accommodate this with only modest modification (taller walls, stronger motor). The observed difference in knee angle etc. are likely evidence that there are still some effect of the sand displacing and dissipating energy, which I predict will be much more obvious if the depth is large enough to remove the solid boundary effect.

2. The author neglects very relevant previous work, especially those from Dan Goldman's lab (e.g., Li, Hsieh, Goldman, 2012, JEB, Qian et al, 2015, B&B) as well as others (e.g., Zhang et al, 2018, PLOS One). Even though the study here is unique in that it concerns bipedal hopping, the main ideas presented in this manuscript (e.g., similar gross kinematics and center of mass oscillation patterns, energy dissipation, compensation by extending legs more to do more muscle work) have been well worked out in these previous studies through more thorough experiments and modeling. This is especially concerning because running as studied in these previous works have observed essentially the same SLIP-like CoM dynamics which hopping (at least on rigid ground) also follows. As good scholars (specifically here but also in general), the authors need to acknowledge previous contributions and strive to carry out their work (both experimentally and during analysis and modeling) to demonstrate how it provides new insight in the context, rather than ignore some of the most relevant literature in order to make it appear like the work is brand new in arriving at these ideas.

Considering this, I strongly encourage the authors to analyze their entire dataset at five speeds to see if some real novel observations and insights remain to be discovered. Another suggestion is evaluating how kinematics changes with speed during the non-steady phases of the trials on sand. The authors mentioned that these recorded trials are not always in steady state due the sand being accelerated. It would be interesting to test if the animal's kinematics shift during this transient and infer what they may be doing in terms of muscular control and force generation, in the context of previous literature.

3. There are a number of vague statements and assumptions that need to be better explained/justified, without which the claims are undermined. To name a few:
- The authors repeatedly state they chose speeds that yield the "most consistent hopping". Please quantify what you mean.
 - Line 131: some hops – how many? Is it the majority of the hops of these trials? If so then this contradicts the argument that the treadmill address the non-steady locomotion previous work suffers from.
 - Line 133: have you looked at the videos carefully (even try manual tracking) to verify the assumption of foot speed being equal to VTRT speed is valid? Previous works (e.g. ones from Goldman's group) have found substantial slippage of foot during running on sand.
 - Line 137: Can you show the data to show that they agree?

4. The authors did not seem to have taken reviewers' valid concerns as seriously as they should.

- a. Throughout the manuscript and response letter, there are many places where the authors refer to data not shown or “unpublished data” to justify their statements. They should add supplementary material or at least include them in response to reviews.
- b. There are also many places where the authors simply address concerns by stating “we prefer not to” without giving convincing arguments.
- c. Just because a journal imposes word limit does not mean one should not address critical concerns. There is quite a bit of repetition of ideas throughout (treadmill limitation, mechanical constraint by sand, to name a few). This is substantial considering how short the manuscript already is. The authors should remove repetition and instead address concerns.

Minor:

I do not agree with the last sentence of abstract. Even without considering the potential flaw in using sand too shallow, the mere kinematic resemblance of a crouched posture does not suggest similar mechanical adaptations. On sand the crouched posture is more of an adaptation for legs to extend and push harder to do work to compensate for the energy loss into sand (which will likely be more obvious if the authors used deeper sand). By contrast, on rough terrain, I suspect it would be more of an adaptation to maintain dynamic stability.

Line 46: Incorrect. Even on rigid ground, legs do lose energy when colliding with the surface.

“Angle of stability” should be “angle of repose”. Also, the authors do not seem to understand this well. A higher angle of repose results from particles having larger particle-particle friction, which means that the bulk of granular media is stronger in withstanding external load. So it is only when a granular media is oriented at a slope close to the angle of repose that it becomes very weak and avalanches easily. In Line

Line 99: What do you mean by “calibrated for spatial measurements”? Only one camera was used so there is no 3-D calibration. What is the lens focal length used?

What is the stride frequency of the animal? Is a 25 Hz cut off frequency sufficient for filtering?

What are the statistical tests being used exactly?

Line 163: The authors assumed that hopping is SLIP-like here. This is not necessarily true. It leaves the impression that the authors were looking for certain features in the data to “prove” something they assumed. What if the hip is not close to the CoM? Also, I don’t quite understand how the hip joint angles alone show the SLIP-like dynamics. Please elaborate.

Line 167-169: Can you show these data?

Fig. 1. It’d be useful to add definition of coordinate system and define EL, length, y length, and the angles being plotted in Figs. 1 and 2. Why not plot a line and band (s.d.) to show take off moment data?

Fig. 2. Need to explain error bars are s.d.

I believe by “cumulative errors”, Reviewer 1 means errors accumulated spatially, not temporally. This still needs to be addressed.

Decision letter (RSOS-201796.R0)

Dear Dr Lin

The Editors assigned to your paper RSOS-201796 "Comparison between the kinematics for kangaroo rat hopping on a solid versus sand surface" have made a decision based on their reading of the paper and any comments received from reviewers.

Regrettably, in view of the reports received, the manuscript has been rejected in its current form. However, a new manuscript may be submitted which takes into consideration these comments.

We invite you to respond to the comments supplied below and prepare a resubmission of your manuscript. Below the referees' and Editors' comments (where applicable) we provide additional requirements. We provide guidance below to help you prepare your revision.

Please note that resubmitting your manuscript does not guarantee eventual acceptance, and we do not generally allow multiple rounds of revision and resubmission, so we urge you to make every effort to fully address all of the comments at this stage. If deemed necessary by the Editors, your manuscript will be sent back to one or more of the original reviewers for assessment. If the original reviewers are not available, we may invite new reviewers.

Please resubmit your revised manuscript and required files (see below) no later than 23-May-2021. Note: the ScholarOne system will 'lock' if resubmission is attempted on or after this deadline. If you do not think you will be able to meet this deadline, please contact the editorial office immediately.

Please note article processing charges apply to papers accepted for publication in Royal Society Open Science (<https://royalsocietypublishing.org/rsos/charges>). Charges will also apply to papers transferred to the journal from other Royal Society Publishing journals, as well as papers submitted as part of our collaboration with the Royal Society of Chemistry (<https://royalsocietypublishing.org/rsos/chemistry>). Fee waivers are available but must be requested when you submit your manuscript (<https://royalsocietypublishing.org/rsos/waivers>).

Thank you for submitting your manuscript to Royal Society Open Science and we look forward to receiving your resubmission. If you have any questions at all, please do not hesitate to get in touch.

on behalf of Dr Monica Daley (Associate Editor) and Kevin Padian (Subject Editor)
openscience@royalsociety.org

Editor comments:

Thank you for your submission. I agree with the AE that some major revisions are needed but also that you may not be able to respond to all of them due to the nature of your experimental approach. Please qualify whatever conclusions are necessary as you see fit; we will likely return it for further review to the referee who had concerns. Best wishes for your revisions.

Associate Editor Comments to Author (Dr Monica Daley):

Associate Editor: 1

Comments to the Author:

Your paper has been reviewed by two experts, who have considered the current manuscript draft as well as the revisions and response to the reviews of the previous Biology Letters version.

Reviewer 1 raises fairly minor comments to address; however, Reviewer 2 raises substantial concerns about the potential for boundary effects in the shallow sand substrate, putting the work into the context of relevant previous literature, and providing a more thorough analysis and discussion that fully addresses the valid concerns raised in previous reviews of the work.

Considering these comments, I cannot accept the paper in its current form, but I will consider a substantially revised and resubmitted version if the authors can fully address the reviewers' concerns about the analysis and interpretation of the work.

While the authors may not be able to completely address all of the points raised, it should be possible to provide substantive additional analysis, explanation, and discussion in the context of relevant previous literature, keeping in mind that RSOS does not have the same space restrictions as Biology Letters.

Reviewer comments to Author:

Reviewer: 1

Comments to the Author(s)

The authors have generally responded well to the reviews received when their MS was submitted to Biology Letters and have clarified several matters that were not clearly presented when first submitted. They have also provided a reasonable argument in response to the reviews for the limited kinematics analysis that they perform, in the absence of ground force recordings (which would be quite difficult with their setup), EMGs of muscles and any in vivo muscle forces. They show that basic very general kinematic patterns of the hindlimbs are not significantly different hopping on sand vs a solid substrate. I do think, however, that they need to quantify EL length at mid-support for both condition in the context of arguing that the limb appears more flexed. This may be the case at landing, but the joint angle patterns shown in Fig. 1C do not support this being the case over much of the support phase. Improved explanation and discussion of this is the only main issue I have regarding the revised MS and the changes made by the authors regarding their findings.

Specific comments:

l. 56-58 "In this previous study on lizards, the locomotion was performed in bursts, implying that the movements may not have been during steady locomotion, thus influencing the kinematic patterns."

Suggest rephrase as "However, locomotion was performed in burst in this previous study of lizards, implying that the movements may not have been during steady locomotion, thus influencing their kinematic patterns."

l. 61-61 "Because of the similarity of the task in this previous study to the impact of landing and pushoff during animal hopping"

Suggest rephrase as: "Because of the similarity of this robotic task to the impact of landing and pushoff during animal hopping"

l. 63-64: last new sentence - this would seem better made as the topic sentence of the paragraph not as its conclusion.

l. 70 suggest: "maintain similar general"

l. 84 "metatarsophalangeal"

l.99-102 "and calibrated for spatial measurements" repeated twice in these sentences

l.160 "patterns" pl. not singular

l.166 horizontal and vertical velocities of ?? (hip as proxy for COM, I believe)

l.168: ditto - presumably these are for hip as COM estimate

l.173 "gross kinematics" add "that we measured"

l.174-177: the authors state that the limb appears more flexed hopping on sand, but the joint angles shown in Fig. 1C don't clearly demonstrate this. To quantify this, they report EL length "at landing". However, this only reflects how flexed the joints may be at landing; not over the course of stance or mid-support when the greatest GRF would be expected to act. The authors should also test for whether EL length differs at mid support to better evaluate the the limb is more flexed in relation to greater joint loading (rather than just a landing and takeoff). Clearer explanation and rationale for these statements and what is shown in Fig. 1C is therefore needed.

l.188 rephrase as: "we presume that hopping on sand imposed"

l.194-195 suggest rephrase as "Thus, it is likely that some neuromuscular compensation occurred to offset..."

l.196 - break into 2 sentences. "substrate. This hypothesis..."
Also "in vivo force transducers" force of what?? (VAGUE)

l.208-210 re: centrifugal force being 1/3 body weight - I would suggest this is a good place to acknowledge limitations of 2D kinematics analysis and that 3D kinematics would help to address any ML compensation by the animals.

Figures

Sample sizes should be given for the plots in 1C and Fig, 2A,B&C. In Figs 1C & 2C, it would help to identify the stance vs swing phases, even though 'takeoff' is labeled. Also, I don't find the light orange - brownish trace for sand very clear - if made darker or a brighter color it would show up better.

Reviewer: 2

Comments to the Author(s)

This study uses a rotary treadmill and high speed imaging to record and compare the kinematics of kangaroo rats hopping on sand vs. solid surfaces. The authors found that the gross kinematic variables are similar, but a few leg kinematic features are different.

The study is novel in that it is the first to compare bipedal hopping kinematics on sand vs. solid surfaces. However, it suffers several major issues that must be addressed before further consideration.

1. It is not clear if the very shallow sand bed tested really behaves like sand or has substantial boundary effect from the bottom. The kangaroo rat's leg length is on the order of 10 cm (from Fig. 2C), and its foot alone is probably 3 cm long. By contrast, the sand bed is merely 2.5 cm deep. It will not be surprising if the toes can reach the rigid bottom, if the foot undergoes substantial rotation subsurface (see Li, Hsieh, Goldman, 2012, JEB, Fig. 11C). Even if it does not, there can be substantial forces from the solid boundary. This is a well-known effect in confined granular media. Even when an intruding object is still at a substantial distance from the walls that contain granular media, forces already start to go up substantially (e.g., see Li, Zhang, Goldman, 2013, Science, Fig. 2B and Materials & Methods first section). My guess would be that the near solid bottom effect due to the very shallow depth of the sand largely contributes to the similar overall kinematics observed. Why not try it again with deeper sand (~10 cm or above)? It looks like the treadmill can well accommodate this with only modest modification (taller walls, stronger motor). The observed difference in knee angle etc. are likely evidence that there are still some effect of the sand displacing and dissipating energy, which I predict will be much more obvious if the depth is large enough to remove the solid boundary effect.

2. The author neglects very relevant previous work, especially those from Dan Goldman's lab (e.g., Li, Hsieh, Goldman, 2012, JEB, Qian et al, 2015, B&B) as well as others (e.g., Zhang et al, 2018, PLOS One). Even though the study here is unique in that it concerns bipedal hopping, the main ideas presented in this manuscript (e.g., similar gross kinematics and center of mass oscillation patterns, energy dissipation, compensation by extending legs more to do more muscle work) have been well worked out in these previous studies through more thorough experiments and modeling. This is especially concerning because running as studied in these previous works have observed essentially the same SLIP-like CoM dynamics which hopping (at least on rigid ground) also follows. As good scholars (specifically here but also in general), the authors need to acknowledge previous contributions and strive to carry out their work (both experimentally and during analysis and modeling) to demonstrate how it provides new insight in the context, rather than ignore some of the most relevant literature in order to make it appear like the work is brand new in arriving at these ideas.

Considering this, I strongly encourage the authors to analyze their entire dataset at five speeds to see if some real novel observations and insights remain to be discovered. Another suggestion is evaluating how kinematics changes with speed during the non-steady phases of the trials on sand. The authors mentioned that these recorded trials are not always in steady state due the sand being accelerated. It would be interesting to test if the animal's kinematics shift during this transient and infer what they may be doing in terms of muscular control and force generation, in the context of previous literature.

3. There are a number of vague statements and assumptions that need to be better explained/justified, without which the claims are undermined. To name a few:

- a. The authors repeatedly state they chose speeds that yield the "most consistent hopping". Please quantify what you mean.
- b. Line 131: some hops—how many? Is it the majority of the hops of these trials? If so then this contradicts the argument that the treadmill address the non-steady locomotion previous work suffers from.
- c. Line 133: have you looked at the videos carefully (even try manual tracking) to verify the assumption of foot speed being equal to VTRT speed is valid? Previous works (e.g. ones from Goldman's group) have found substantial slippage of foot during running on sand.
- d. Line 137: Can you show the data to show that they agree?

4. The authors did not seem to have taken reviewers' valid concerns as seriously as they should.
- Throughout the manuscript and response letter, there are many places where the authors refer to data not shown or "unpublished data" to justify their statements. They should add supplementary material or at least include them in response to reviews.
 - There are also many places where the authors simply address concerns by stating "we prefer not to" without giving convincing arguments.
 - Just because a journal imposes word limit does not mean one should not address critical concerns. There is quite a bit of repetition of ideas throughout (treadmill limitation, mechanical constraint by sand, to name a few). This is substantial considering how short the manuscript already is. The authors should remove repetition and instead address concerns.

Minor:

I do not agree with the last sentence of abstract. Even without considering the potential flaw in using sand too shallow, the mere kinematic resemblance of a crouched posture does not suggest similar mechanical adaptations. On sand the crouched posture is more of an adaptation for legs to extend and push harder to do work to compensate for the energy loss into sand (which will likely be more obvious if the authors used deeper sand). By contrast, on rough terrain, I suspect it would be more of an adaptation to maintain dynamic stability.

Line 46: Incorrect. Even on rigid ground, legs do lose energy when colliding with the surface.

"Angle of stability" should be "angle of repose". Also, the authors do not seem to understand this well. A higher angle of repose results from particles having larger particle-particle friction, which means that the bulk of granular media is stronger in withstanding external load. So It is only when a granular media is oriented at a slope close to the angle of repose that it becomes very weak and avalanches easily. In Line

Line 99: What do you mean by "calibrated for spatial measurements"? Only one camera was used so there is no 3-D calibration. What is the lens focal length used?

What is the stride frequency of the animal? Is a 25 Hz cut off frequency sufficient for filtering?

What are the statistical tests being used exactly?

Line 163: The authors assumed that hopping is SLIP-like here. This is not necessarily true. It leaves the impression that the authors were looking for certain features in the data to "prove" something they assumed. What if the hip is not close to the CoM? Also, I don't quite understand how the hip joint angles alone show the SLIP-like dynamics. Please elaborate.

Line 167-169: Can you show these data?

Fig. 1. It'd be useful to add definition of coordinate system and define EL, length, y length, and the angles being plotted in Figs. 1 and 2. Why not plot a line and band (s.d.) to show take off moment data?

Fig. 2. Need to explain error bars are s.d.

I believe by "cumulative errors", Reviewer 1 means errors accumulated spatially, not temporally. This still needs to be addressed.

===PREPARING YOUR MANUSCRIPT===

===PREPARING YOUR REVISION IN SCHOLARONE===

- An individual file of each figure (EPS or print-quality PDF preferred [either format should be produced directly from original creation package], or original software format).
- An editable file of each table (.doc, .docx, .xls, .xlsx, or .csv).
- An editable file of all figure and table captions.

- Any electronic supplementary material (ESM).
- If you are requesting a discretionary waiver for the article processing charge, the waiver form must be included at this step.
- If you are providing image files for potential cover images, please upload these at this step, and inform the editorial office you have done so. You must hold the copyright to any image provided.
- A copy of your point-by-point response to referees and Editors. This will expedite the preparation of your proof.

- Ensure that your data access statement meets the requirements at <https://royalsociety.org/journals/authors/author-guidelines/#data>. You should ensure that you cite the dataset in your reference list. If you have deposited data etc in the Dryad repository, please include both the 'For publication' link and 'For review' link at this stage.
- If you are requesting an article processing charge waiver, you must select the relevant waiver option (if requesting a discretionary waiver, the form should have been uploaded at Step 3 'File upload' above).
- If you have uploaded ESM files, please ensure you follow the guidance at <https://royalsociety.org/journals/authors/author-guidelines/#supplementary-material> to include a suitable title and informative caption. An example of appropriate titling and captioning may be found at https://figshare.com/articles/Table_S2_from_Is_there_a_trade-off_between_peak_performance_and_performance_breadth_across_temperatures_for_aerobic_scope_in_teleost_fishes_/3843624.

Author's Response to Decision Letter for (RSOS-201796.R0)

See Appendix A.

RSOS-211491.R0

Review form: Reviewer 1

Is the manuscript scientifically sound in its present form?

Yes

Are the interpretations and conclusions justified by the results?

Yes

Is the language acceptable?

Yes

Do you have any ethical concerns with this paper?

No

Have you any concerns about statistical analyses in this paper?

No

Recommendation?

Accept as is

Comments to the Author(s)

The authors have responded well to my more modest concerns. R2 brought up several good points, the main ones being (1) whether the sand depth used (2.5 cm) was sufficient to avoid boundary effects due to granular compaction relative to the wood surface, and (2) the need to provide broader context for their study of kangaroo rats hopping on sand relative to other studies that have examined locomotion of other species over granular substrates. They now include a very useful discussion in relation to these other studies (which Roy Soc Open allows). The authors also have carried out appropriate penetration resistance experiments that show beyond 1.5 cm there is no statistical effect. This additional analysis (Fig. 4) is an important addition to their paper.

The other concerns raised by R2 are ones that the authors' data set and the kangaroo rats' behavioral response to the rotating 'treadmill' really cannot address well. In general, the authors have addressed these concerns and qualified their results appropriately.

This is now an improved and stronger paper that makes a valuable contribution to the literature on how animals move over granular vs solid substrates, with relevance to their ecology and how the foot morphology of kangaroo rats provides evidence for adaptation to moving effectively over a sand substrate.

Review form: Reviewer 2

Is the manuscript scientifically sound in its present form?

No

Are the interpretations and conclusions justified by the results?

No

Is the language acceptable?

Yes

Do you have any ethical concerns with this paper?

No

Have you any concerns about statistical analyses in this paper?

No

Recommendation?

Major revision is needed (please make suggestions in comments)

Comments to the Author(s)

The authors have done a good job addressing much of the major comments and most of the minor comments. I am encouraged by their attempt to address the biggest potential issue of boundary effect. The author's addition of granular force measurements to test the boundary effect and related discussion is encouraging in terms of scholarship and demonstrates that the authors are making a real effort to understand the physics of penetration into granular media, apply it to their analysis, and discuss their observation in this context. However, I do have some important questions about this experiment, due to which I remain not convinced that boundary effect is not present. I do believe that, even if this issue is true as I suspect, it is simply because the authors are still relatively new to granular penetration physics for which there are such challenging issues that can take time and practice to fully appreciate.

Major:

1. The authors still simply motivated their work by stating that hopping on different substrates is not well studied. This is true, but there are many questions not well studied in the world. In other words, this simple justification can be stated for most questions, and it is not compelling for readers why he/she should care. What makes this important and useful? This remains not explained.

In the same spirit, the authors still have not fully discussed similarities and differences from Li et al. 2012 JEB, which observed or speculated many similar things as I pointed out in the previous review (e.g., similar gross kinematics and center of mass oscillation patterns, energy dissipation, compensation by extending legs more to do more muscle work).

2. What still puzzled me is that the measured effective stiffness (the slope of force vs. penetration depth or "penetration resistance" in Fig. 4) is extremely high. Even for the sand bed of larger depths where the authors believe there is no boundary effect, the slope is merely 2-3 times lower than without sand (0 sand bed depth, blue curve), which is pushing against wood. This is very surprising, because granular media that is deep enough to not have boundary effect should have orders of magnitude lower effective stiffness than a solid such as wood.

For example, for the model granular media studied in Li et al 2013 Science, the penetration resistance of the foot tested here (2.5 cm × 1.25 cm) would range from 0.2 to 0.5 N/cm³ × 2.5 cm × 1.25 cm = 0.625 to 1.5625 N/cm. This means that two horizontal kangaroo rat feet loaded at 3 N assumed here would penetrate to 0.5 × 3 N / 1.5625 N/cm to 0.5 × 3 N / 0.625 N/cm = 0.96 to 2.4 cm deep. This would correspond to 10-50% of the effective leg length (Fig. 3A). This is further evidenced in the very small leg penetration ratio of 0.0056 (less than 1% of the leg penetrating) vs. the much larger values for the diversity of lizards, geckos, and crab studied in Qian et al 2015. I find it extremely surprising that the kangaroo rat with an effective leg length of 4-8 cm would penetrate only less than 1 mm into natural sand, which is probably much less than even its toe thickness.

By contrast, the authors measured a very high penetration resistance of 3 N / 0.7 N/mm = 43 N/cm, almost two orders of magnitude greater than the granular media studied in Li et al. 2013. This is very surprising. Although natural sand has polydispersity and angularity that will increase mechanical strength, the increase is typically well within an order of magnitude (see Fig. S11 of Li et al 2013). For example, the strongest sand (29 Palms sand) from the Mojave desert that Li et al 2013 studied has a penetration resistance of 2.5 N/cm² / 3 cm × 2.5 cm × 1.25 cm = 2.6 N/cm, only less than twice that of the highest of the model granular media (0.625 to 1.5625 N/cm).

Because of all these, I still strongly suspect that the sand bed depth is too small and the penetrating object is feeling a very large force, much of which is actually from a much stronger granular media due to the boundary effect than the natural sand that has a very large depth.

I strongly suggest the authors redo their force measurements with a very deep sand bed (10s of cm) to check this.

If this indeed was the case, I do strongly encourage the authors to redo the sand hopping experiments with a deep sand bed. I appreciate that this may be much more work than the authors may have planned to do for this study, but not having boundary effect is a key premise of almost all the hypotheses and interpretations of the study.

In the same spirit, after checking the above, I suggest that the authors consider foot rotation. You can do a simple calculation: assuming that the foot is a plate rotating about the ankle along an axis that is lateral (horizontal), then the foot pressure is going to linearly increase from the ankle to the toe (as was done in Li et al. 2012 JEB). The average pressure is simply the pressure at the middle. Thus you can figure out how deep the toe would reach.

3. Some other important technical details of the new force measurement experiments are not provided. These include: what force sensor is used, how force is limited to 3 N during penetration, and how exactly is the motor controlled.

In particular, one technical question is that the motor is controlled to generate a high linear speed of 50 cm/s. However, the motor is stopped when force reaches 3 N, which, depending on the total depth of sand used, is only a displacement of 0.2 to 0.7 mm. This is a tiny distance to cover even at a high speed of 50 cm/s (500 mm/s). This leads me to suspect that the motor barely has enough time to accelerate to the desired 50 cm/s. In fact, even at a slow speed, a displacement of less than 1 mm is small for a servo to generate precisely, unless there is a large gear ratio.

Can you provide a video (ideally high speed) to show this experiment so I can better understand how the penetration motion is generated?

It would also be useful to add a schematics to show this experimental setup. In particular, the authors should clearly define sand bed so that it is easy to differentiate it from penetration depth discussed later and used in previous studies. The authors should also better refer to Fig. 3 elements when describing experimental methods and results.

Minor:

1. Line 49: Make it clear this is mechanical work, not metabolic work.
2. Line 78: It would be more useful to explain the mechanism (that increase in foot pressure leads to deeper leg penetration which leads to more energy dissipation), rather than simply "combination of several factors". Even though many biological phenomena are complex and lack causal understanding, in this case, previous work has well explained the mechanisms (especially Li et al. 2012 JEB and Qian 2015 B&B). The same comment goes for "interact to affect..." in Line 311, and "all influence..." in Line 297.
3. Line 140: So 0% of hard ground trials? May be worth mentioning.

4. I still don't understand what "most valid" means. Most in what sense? For scientific writing, words like most/more is not very meaningful without quantification or at least a definition, as it can be arbitrary.
5. It would be useful for the authors to briefly explain how the angle of stability and bulk density they used here differs from and are related to angle of repose and material density (which multiplied by volume fraction is bulk density). This would be useful for readers not familiar with granular media, as those other terms are used in some of the closely relevant sand locomotion studies.
6. Line 177: A servo motor directly outputs a rotation. Explain how this is converted to linear velocity.
7. In general, I find short acronyms like PR, EL easier to simply spell out. It can be a bit hard to remember at times.
8. Line 334: Why is it potentially more important?
9. Fig. 1 has a low resolution. The new caption text seems to belong to Methods.
10. Fig. 2 caption. Can you also add total trial number?
11. I feel the author's response to my previous comments on different speed and non-steady phases trials are informative. It may be useful to mention some of it.
12. The author's response "We have added a reference to support this statement for hopping." Which reference exactly?

Decision letter (RSOS-211491.R0)

Dear Dr Lin

On behalf of the Editors, we are pleased to inform you that your Manuscript RSOS-211491 "Comparison between the kinematics for kangaroo rat hopping on a solid versus sand surface" has been accepted for publication in Royal Society Open Science subject to minor revision in accordance with the referees' reports. Please find the referees' comments along with any feedback from the Editors below my signature.

Please submit your revised manuscript and required files (see below) no later than 7 days from today's (ie 22-Nov-2021) date. Note: the ScholarOne system will 'lock' if submission of the revision is attempted 7 or more days after the deadline. If you do not think you will be able to meet this deadline please contact the editorial office immediately.

on behalf of Dr Jonas Rubenson (Associate Editor) and Kevin Padian (Subject Editor)
openscience@royalsociety.org

Associate Editor Comments to Author (Dr Jonas Rubenson):

Associate Editor

Comments to the Author:

Dear Dr. Lin,

As you will see, the reviews of your revised manuscript are mixed. R1 is satisfied with how you addressed the reviewer critiques, including your new experiments to address possible boundary effects of foot-sand penetration. I agree that the new experiments have gone a long way to address the original issues raised in the first review. R2, on the other hand, raises additional concerns about your new boundary effect experiments.

R2 has diligently provided a thorough comparison of your results to previous mechanical studies into granular media. R2 points out that your data is different from these previous studies. An important distinction, however, between these previous studies and the new experiment presented here is the sand penetration velocity. In those previous studies, it appears the velocity was 1 cm/s. In contrast, in this study a velocity of 50cm/s was used, replicating the velocity of the kangaroo rat foot. The inertial effect of the sand motion makes a direct comparison between studies complicated.

Although you and your co-authors could replicate the force experiments with deeper sand (10cm +), in this case I do not think it is essential. The force experiments up to 5cm depth appear to be performed well using an approach that mimics the animal foot mechanics. Comparison to previous data does not, by itself, invalidate these experiments, nor the locomotor studies performed. Furthermore, it is not clear why we would expect to see an initial large boundary effect, a stable region, followed by a new boundary effect with deeper sand. I nevertheless urge you and your co-authors to discuss the differences in the penetration force data to previous literature, and also to highlight how the experimental set-up may differ from the natural sand environment. R2 also requests specific additional information regarding the experimental set-up, as well as other relevant comments, that I recommend you provide/address.

Best Regards,
Jonas Rubenson

Reviewer comments to Author:

Reviewer: 1

Comments to the Author(s)

The authors have responded well to my more modest concerns. R2 brought up several good points, the main ones being (1) whether the sand depth used (2.5 cm) was sufficient to avoid boundary effects due to granular compaction relative to the wood surface, and (2) the need to provide broader context for their study of kangaroo rats hopping on sand relative to other studies that have examined locomotion of other species over granular substrates. They now include a very useful discussion in relation to these other studies (which Roy Soc Open allows). The authors also have carried out appropriate penetration resistance experiments that show beyond 1.5 cm there is no statistical effect. This additional analysis (Fig. 4) is an important addition to their paper.

The other concerns raised by R2 are ones that the authors' data set and the kangaroo rats' behavioral response to the rotating 'treadmill' really cannot address well. In general, the authors have addressed these concerns and qualified their results appropriately.

This is now an improved and stronger paper that makes a valuable contribution to the literature on how animals move over granular vs solid substrates, with relevance to their ecology and how the foot morphology of kangaroo rats provides evidence for adaptation to moving effectively over a sand substrate.

Reviewer: 2

Comments to the Author(s)

The authors have done a good job addressing much of the major comments and most of the minor comments. I am encouraged by their attempt to address the biggest potential issue of boundary effect. The author's addition of granular force measurements to test the boundary effect and related discussion is encouraging in terms of scholarship and demonstrates that the authors are making a real effort to understand the physics of penetration into granular media, apply it to their analysis, and discuss their observation in this context. However, I do have some important questions about this experiment, due to which I remain not convinced that boundary effect is not present. I do believe that, even if this issue is true as I suspect, it is simply because the authors are still relatively new to granular penetration physics for which there are such challenging issues that can take time and practice to fully appreciate.

Major:

1. The authors still simply motivated their work by stating that hopping on different substrates is not well studied. This is true, but there are many questions not well studied in the world. In other words, this simple justification can be stated for most questions, and it is not compelling for readers why he/she should care. What makes this important and useful? This remains not explained.

In the same spirit, the authors still have not fully discussed similarities and differences from Li et al. 2012 JEB, which observed or speculated many similar things as I pointed out in the previous review (e.g., similar gross kinematics and center of mass oscillation patterns, energy dissipation, compensation by extending legs more to do more muscle work).

2. What still puzzled me is that the measured effective stiffness (the slope of force vs. penetration depth or "penetration resistance" in Fig. 4) is extremely high. Even for the sand bed of larger depths where the authors believe there is no boundary effect, the slope is merely 2-3 times lower

than without sand (0 sand bed depth, blue curve), which is pushing against wood. This is very surprising, because granular media that is deep enough to not have boundary effect should have orders of magnitude lower effective stiffness than a solid such as wood.

For example, for the model granular media studied in Li et al 2013 Science, the penetration resistance of the foot tested here (2.5 cm × 1.25 cm) would range from 0.2 to 0.5 N/cm³ × 2.5 cm × 1.25 cm = 0.625 to 1.5625 N/cm. This means that two horizontal kangaroo rat feet loaded at 3 N assumed here would penetrate to 0.5 × 3 N / 1.5625 N/cm to 0.5 × 3 N / 0.625 N/cm = 0.96 to 2.4 cm deep. This would correspond to 10-50% of the effective leg length (Fig. 3A). This is further evidenced in the very small leg penetration ratio of 0.0056 (less than 1% of the leg penetrating) vs. the much larger values for the diversity of lizards, geckos, and crab studied in Qian et al 2015. I find it extremely surprising that the kangaroo rat with an effective leg length of 4-8 cm would penetrate only less than 1 mm into natural sand, which is probably much less than even its toe thickness.

By contrast, the authors measured a very high penetration resistance of 3 N / 0.7 N/mm = 43 N/cm, almost two orders of magnitude greater than the granular media studied in Li et al. 2013. This is very surprising. Although natural sand has polydispersity and angularity that will increase mechanical strength, the increase is typically well within an order of magnitude (see Fig. S11 of Li et al 2013). For example, the strongest sand (29 Palms sand) from the Mojave desert that Li et al 2013 studied has a penetration resistance of 2.5 N/cm² / 3 cm × 2.5 cm × 1.25 cm = 2.6 N/cm, only less than twice that of the highest of the model granular media (0.625 to 1.5625 N/cm).

Because of all these, I still strongly suspect that the sand bed depth is too small and the penetrating object is feeling a very large force, much of which is actually from a much stronger granular media due to the boundary effect than the natural sand that has a very large depth.

I strongly suggest the authors redo their force measurements with a very deep sand bed (10s of cm) to check this.

If this indeed was the case, I do strongly encourage the authors to redo the sand hopping experiments with a deep sand bed. I appreciate that this may be much more work than the authors may have planned to do for this study, but not having boundary effect is a key premise of almost all the hypotheses and interpretations of the study.

In the same spirit, after checking the above, I suggest that the authors consider foot rotation. You can do a simple calculation: assuming that the foot is a plate rotating about the ankle along an axis that is lateral (horizontal), then the foot pressure is going to linearly increase from the ankle to the toe (as was done in Li et al. 2012 JEB). The average pressure is simply the pressure at the middle. Thus you can figure out how deep the toe would reach.

3. Some other important technical details of the new force measurement experiments are not provided. These include: what force sensor is used, how force is limited to 3 N during penetration, and how exactly is the motor controlled.

In particular, one technical question is that the motor is controlled to generate a high linear speed of 50 cm/s. However, the motor is stopped when force reaches 3 N, which, depending on the total depth of sand used, is only a displacement of 0.2 to 0.7 mm. This is a tiny distance to cover even at a high speed of 50 cm/s (500 mm/s). This leads me to suspect that the motor barely has enough time to accelerate to the desired 50 cm/s. In fact, even at a slow speed, a displacement of less than 1 mm is small for a servo to generate precisely, unless there is a large gear ratio.

Can you provide a video (ideally high speed) to show this experiment so I can better understand how the penetration motion is generated?

It would also be useful to add a schematics to show this experimental setup. In particular, the authors should clearly define sand bed so that it is easy to differentiate it from penetration depth discussed later and used in previous studies. The authors should also better refer to Fig. 3 elements when describing experimental methods and results.

Minor:

1. Line 49: Make it clear this is mechanical work, not metabolic work.
2. Line 78: It would be more useful to explain the mechanism (that increase in foot pressure leads to deeper leg penetration which leads to more energy dissipation), rather than simply "combination of several factors". Even though many biological phenomena are complex and lack causal understanding, in this case, previous work has well explained the mechanisms (especially Li et al. 2012 JEB and Qian 2015 B&B). The same comment goes for "interact to affect..." in Line 311, and "all influence..." in Line 297.
3. Line 140: So 0% of hard ground trials? May be worth mentioning.
4. I still don't understand what "most valid" means. Most in what sense? For scientific writing, words like most/more is not very meaningful without quantification or at least a definition, as it can be arbitrary.
5. It would be useful for the authors to briefly explain how the angle of stability and bulk density they used here differs from and are related to angle of repose and material density (which multiplied by volume fraction is bulk density). This would be useful for readers not familiar with granular media, as those other terms are used in some of the closely relevant sand locomotion studies.
6. Line 177: A servo motor directly outputs a rotation. Explain how this is converted to linear velocity.
7. In general, I find short acronyms like PR, EL easier to simply spell out. It can be a bit hard to remember at times.
8. Line 334: Why is it potentially more important?
9. Fig. 1 has a low resolution. The new caption text seems to belong to Methods.
10. Fig. 2 caption. Can you also add total trial number?
11. I feel the author's response to my previous comments on different speed and non-steady phases trials are informative. It may be useful to mention some of it.
12. The author's response "We have added a reference to support this statement for hopping." Which reference exactly?

===PREPARING YOUR MANUSCRIPT===

one version should clearly identify all the changes that have been made (for instance, in coloured highlight, in bold text, or tracked changes);

===PREPARING YOUR REVISION IN SCHOLARONE===

- An individual file of each figure (EPS or print-quality PDF preferred [either format should be produced directly from original creation package], or original software format).
- An editable file of each table (.doc, .docx, .xls, .xlsx, or .csv).
- An editable file of all figure and table captions.

- Any electronic supplementary material (ESM).
- If you are requesting a discretionary waiver for the article processing charge, the waiver form must be included at this step.
- If you are providing image files for potential cover images, please upload these at this step, and inform the editorial office you have done so. You must hold the copyright to any image provided.
- A copy of your point-by-point response to referees and Editors. This will expedite the preparation of your proof.

- Ensure that your data access statement meets the requirements at <https://royalsociety.org/journals/authors/author-guidelines/#data>. You should ensure that you cite the dataset in your reference list. If you have deposited data etc in the Dryad repository, please only include the 'For publication' link at this stage. You should remove the 'For review' link.
- If you are requesting an article processing charge waiver, you must select the relevant waiver option (if requesting a discretionary waiver, the form should have been uploaded, see 'File upload' above).
- If you have uploaded any electronic supplementary (ESM) files, please ensure you follow the guidance at <https://royalsociety.org/journals/authors/author-guidelines/#supplementary-material> to include a suitable title and informative caption. An example of appropriate titling and captioning may be found at https://figshare.com/articles/Table_S2_from_Is_there_a_trade-off_between_peak_performance_and_performance_breadth_across_temperatures_for_aerobic_scope_in_teleost_fishes_/3843624.

Author's Response to Decision Letter for (RSOS-211491.R0)

See Appendix B.

Decision letter (RSOS-211491.R1)

Dear Dr Lin,

I am pleased to inform you that your manuscript entitled "Comparison between the kinematics for kangaroo rat hopping on a solid versus sand surface" is now accepted for publication in Royal Society Open Science.

on behalf of Dr Jonas Rubenson (Associate Editor) and Kevin Padian (Subject Editor)
openscience@royalsociety.org

Appendix A

Dear Associate Editor and reviewers: Please see our responses to the latest review of the manuscript, which follow the reviewers' original comments and are italicized.

Associate Editor Comments to Author (Dr Jonas Rubenson):

Comments to the Author:

Dear Dr. Lin,

As you will see, the reviews of your revised manuscript are mixed. R1 is satisfied with how you addressed the reviewer critiques, including your new experiments to address possible boundary effects of foot-sand penetration. I agree that the new experiments have gone a long way to address the original issues raised in the first review. R2, on the other hand, raises additional concerns about your new boundary effect experiments.

R2 has diligently provided a thorough comparison of your results to previous mechanical studies into granular media. R2 points out that your data is different from these previous studies. An important distinction, however, between these previous studies and the new experiment presented here is the sand penetration velocity. In those previous studies, it appears the velocity was 1 cm/s. In contrast, in this study a velocity of 50cm/s was used, replicating the velocity of the kangaroo rat foot. The inertial effect of the sand motion makes a direct comparison between studies complicated.

Although you and your co-authors could replicate the force experiments with deeper sand (10cm +), in this case I do not think it is essential. The force experiments up to 5cm depth appear to be performed well using an approach that mimics the animal foot mechanics. Comparison to previous data does not, by itself, invalidate these experiments, nor the locomotor studies performed. Furthermore, it is not clear why we would expect to see an initial large boundary effect, a stable region, followed by a new boundary effect with deeper sand. I nevertheless urge you and your co-authors to discuss the differences in the penetration force data to previous literature, and also to highlight how the experimental set-up may differ from the natural sand environment. R2 also requests specific additional information regarding the experimental set-up, as well as other relevant comments, that I recommend you provide/address.

*We wish to thank the Associate Editor for the notice of acceptance and for clearly stating what revisions are necessary for publication. We have added the three specific recommendations: 1) "discuss the differences in the penetration force data to previous literature" 2) "highlight how the experimental set-up may differ from the natural sand environment" and 3) "specific additional information regarding the experimental set-up, as well as other relevant comments." All the changes and additions to the manuscript are in **red text**.*

Reviewer comments to Author:

Reviewer: 1

Comments to the Author(s)

The authors have responded well to my more modest concerns. R2 brought up several good

points, the main ones being (1) whether the sand depth used (2.5 cm) was sufficient to avoid boundary effects due to granular compaction relative to the wood surface, and (2) the need to provide broader context for their study of kangaroo rats hopping on sand relative to other studies that have examined locomotion of other species over granular substrates. They now include a very useful discussion in relation to these other studies (which Roy Soc Open allows). The authors also have carried out appropriate penetration resistance experiments that show beyond 1.5 cm there is no statistical effect. This additional analysis (Fig. 4) is an important addition to their paper.

The other concerns raised by R2 are ones that the authors' data set and the kangaroo rats' behavioral response to the rotating 'treadmill' really cannot address well. In general, the authors have addressed these concerns and qualified their results appropriately.

This is now an improved and stronger paper that makes a valuable contribution to the literature on how animals move over granular vs solid substrates, with relevance to their ecology and how the foot morphology of kangaroo rats provides evidence for adaptation to moving effectively over a sand substrate.

We thank the reviewer for her/his previous recommendations which substantially improved the paper.

Reviewer: 2

Comments to the Author(s)

The authors have done a good job addressing much of the major comments and most of the minor comments. I am encouraged by their attempt to address the biggest potential issue of boundary effect. The author's addition of granular force measurements to test the boundary effect and related discussion is encouraging in terms of scholarship and demonstrates that the authors are making a real effort to understand the physics of penetration into granular media, apply it to their analysis, and discuss their observation in this context. However, I do have some important questions about this experiment, due to which I remain not convinced that boundary effect is not present. I do believe that, even if this issue is true as I suspect, it is simply because the authors are still relatively new to granular penetration physics for which there are such challenging issues that can take time and practice to fully appreciate.

We thank the reviewer for her/his previous recommendations which substantially improved the paper.

Major:

1. The authors still simply motivated their work by stating that hopping on different substrates is not well studied. This is true, but there are many questions not well studied in the world. In other words, this simple justification can be stated for most questions, and it is not compelling for

readers why he/she should care. What makes this important and useful? This remains not explained.

In the same spirit, the authors still have not fully discussed similarities and differences from Li et al. 2012 JEB, which observed or speculated many similar things as I pointed out in the previous review (e.g., similar gross kinematics and center of mass oscillation patterns, energy dissipation, compensation by extending legs more to do more muscle work).

We appreciate the reviewer's continuing reminder for the need to compare to existing literature. With regard to the comparison to Li et al. (2012), we did reference the work in the Introduction and Discussion but hesitate to perform an in-depth comparison because of the differences between lizard quadrupedal locomotion and the bipedal hopping of kangaroo rats.

2. What still puzzled me is that the measured effective stiffness (the slope of force vs. penetration depth or "penetration resistance" in Fig. 4) is extremely high. Even for the sand bed of larger depths where the authors believe there is no boundary effect, the slope is merely 2-3 times lower than without sand (0 sand bed depth, blue curve), which is pushing against wood. This is very surprising, because granular media that is deep enough to not have boundary effect should have orders of magnitude lower effective stiffness than a solid such as wood.

For example, for the model granular media studied in Li et al 2013 Science, the penetration resistance of the foot tested here ($2.5 \text{ cm} \times 1.25 \text{ cm}$) would range from 0.2 to $0.5 \text{ N/cm}^3 \times 2.5 \text{ cm} \times 1.25 \text{ cm} = 0.625$ to 1.5625 N/cm . This means that two horizontal kangaroo rat feet loaded at 3 N assumed here would penetrate to $0.5 \times 3 \text{ N} / 1.5625 \text{ N/cm}$ to $0.5 \times 3 \text{ N} / 0.625 \text{ N/cm} = 0.96$ to 2.4 cm deep. This would correspond to 10-50% of the effective leg length (Fig. 3A). This is further evidenced in the very small leg penetration ratio of 0.0056 (less than 1% of the leg penetrating) vs. the much larger values for the diversity of lizards, geckos, and crab studied in Qian et al 2015. I find it extremely surprising that the kangaroo rat with an effective leg length of 4-8 cm would penetrate only less than 1 mm into natural sand, which is probably much less than even its toe thickness.

By contrast, the authors measured a very high penetration resistance of $3 \text{ N} / 0.7 \text{ N/mm} = 43 \text{ N/cm}$, almost two orders of magnitude greater than the granular media studied in Li et al. 2013. This is very surprising. Although natural sand has polydispersity and angularity that will increase mechanical strength, the increase is typically well within an order of magnitude (see Fig. S11 of Li et al 2013). For example, the strongest sand (29 Palms sand) from the Mojave desert that Li et al 2013 studied has a penetration resistance of $2.5 \text{ N/cm}^2 / 3 \text{ cm} \times 2.5 \text{ cm} \times 1.25 \text{ cm} = 2.6 \text{ N/cm}$, only less than twice that of the highest of the model granular media (0.625 to 1.5625 N/cm).

Because of all these, I still strongly suspect that the sand bed depth is too small and the penetrating object is feeling a very large force, much of which is actually from a much stronger granular media due to the boundary effect than the natural sand that has a very large depth.

I strongly suggest the authors redo their force measurements with a very deep sand bed (10s of cm) to check this.

If this indeed was the case, I do strongly encourage the authors to redo the sand hopping experiments with a deep sand bed. I appreciate that this may be much more work than the authors may have planned to do for this study, but not having boundary effect is a key premise of almost all the hypotheses and interpretations of the study.

We appreciate the reviewer's further analysis of our penetration resistance data in comparison to existing literature. As the Associate Editor recommended, we included additional text to include this comparison, which emphasizes that differences in penetration velocity hinder direct comparisons (see Associate Editor's message) as well as pointing out that the fluidized setup is different. We also corrected an error in the calculation of \tilde{d} (now equal to 0.0125).

Lines 338-344: We also note that the value of 4.8 N/mm for PR measured in this study is substantially greater than the measured values for sand (29 Palms sand) in a previous study (4), which was approximately 2.6 N/cm when scaled by the area of our penetrator. Major differences in the testing setup between the previous study and our study could be responsible for this discrepancy, namely the speed of the penetrator (50 cm/s (this study) vs 1 cm/s) and the sand being fluidized in the previous study. However, even with using the substantially smaller value of 2.6 N/cm for PR, the value of \tilde{d} is equal to 0.23.

In the same spirit, after checking the above, I suggest that the authors consider foot rotation. You can do a simple calculation: assuming that the foot is a plate rotating about the ankle along an axis that is lateral (horizontal), then the foot pressure is going to linearly increase from the ankle to the toe (as was done in Li et al. 2012 JEB). The average pressure is simply the pressure at the middle. Thus you can figure out how deep the toe would reach.

The amount of foot rotation is an important aspect of the kinematics that we could not measure. While we appreciate the thought experiment suggested by the reviewer, kangaroo rats land and take off on their toes, which would not be well approximated by a solid plate. Moreover, we do have some qualitative observations (not included in the manuscript) that toe rotation was minimal. First, on the single animal, we had an additional overhead camera. We could clearly see the top of the animal's toe nails throughout the landing phase, indicating that their toes were not rotating because otherwise, the nails would have been obscured in sand. Second, kangaroo rats have relatively long (2-4 mm) toenails, and the nails would likely help prevent rotation.

The associate editor also asked to “highlight how the experimental set-up may differ from the natural sand environment” and we address this request by responding to the reviewer's comments,

Lines 344-349: Lastly, we note this analysis is limited by potential differences between our testing method and the natural toe-sand interaction. First, the penetrator was made of solid wood, which will interact with the sand differently than the more flexible kangaroo rat toes. Second, rotation of the toes, if it occurs during hopping, may produce a different penetration effect than

the perpendicular intrusion of the penetrator used. More detailed analyses are needed to address these uncertainties.

3. Some other important technical details of the new force measurement experiments are not provided. These include: what force sensor is used, how force is limited to 3 N during penetration, and how exactly is the motor controlled.

We have included this information in the revised manuscript (lines 177, 183, 185).

In particular, one technical question is that the motor is controlled to generate a high linear speed of 50 cm/s. However, the motor is stopped when force reaches 3 N, which, depending on the total depth of sand used, is only a displacement of 0.2 to 0.7 mm. This is a tiny distance to cover even at a high speed of 50 cm/s (500 mm/s). This leads me to suspect that the motor barely has enough time to accelerate to the desired 50 cm/s. In fact, even at a slow speed, a displacement of less than 1 mm is small for a servo to generate precisely, unless there is a large gear ratio.

The reviewer is correct in that the servo did not have time to reach the commanded speed of 50 cm/s. However, the servo produced motion (position vs time) that was similar in all the conditions, thus the penetration resistance measurement was a result of the condition tested. The fact that the penetration resistance did change for the shallow sand depths supports this conclusion.

Can you provide a video (ideally high speed) to show this experiment so I can better understand how the penetration motion is generated?

It would also be useful to add a schematics to show this experimental setup. In particular, the authors should clearly define sand bed so that it is easy to differentiate it from penetration depth discussed later and used in previous studies. The authors should also better refer to Fig. 3 elements when describing experimental methods and results.

We are uncertain about the request to better reference Fig. 3, which is concerning kinematic analyses. Perhaps, the reviewer meant Fig. 4? We hope that the additional details added in the Methods (see above) will allow the reader to understand the experimental setup more easily. We added text to the Results section to better explain Fig. 4 (lines 244-245).

Minor:

1. Line 49: Make it clear this is mechanical work, not metabolic work.

We have revised the text as suggested.

2. Line 78: It would be more useful to explain the mechanism (that increase in foot pressure leads to deeper leg penetration which leads to more energy dissipation), rather than simply “combination of several factors”. Even though many biological phenomena are complex and lack

causal understanding, in this case, previous work has well explained the mechanisms (especially Li et al. 2012 JEB and Qian 2015 B&B). The same comment goes for “interact to affect...” in Line 311, and “all influence...” in Line 297.

We have revised the text as suggested (lines 79-80).

3. Line 140: So 0% of hard ground trials? May be worth mentioning.

We have revised the text as suggested.

4. I still don't understand what “most valid” means. Most in what sense? For scientific writing, words like most/more is not very meaningful without quantification or at least a definition, as it can be arbitrary.

We did not intend to use “most” as an adjective for “valid.” We revised the text to: “which produced the most hops deemed as valid ...”

5. It would be useful for the authors to briefly explain how the angle of stability and bulk density they used here differs from and are related to angle of repose and material density (which multiplied by volume fraction is bulk density). This would be useful for readers not familiar with granular media, as those other terms are used in some of the closely relevant sand locomotion studies.

We have added text as suggested.

Lines 170-172: Note that angle of stability and bulk density differ from the properties of angle of repose (the steepest angle at which the material can be piled) and material density (bulk density divided by volume fraction), respectively.

6. Line 177: A servo motor directly outputs a rotation. Explain how this is converted to linear velocity.

We have added text as suggested. Lines 180-182.

7. In general, I find short acronyms like PR, EL easier to simply spell out. It can be a bit hard to remember at times.

Because the quantities are measured or calculated variables, we had to provide them with a short acronym.

8. Line 334: Why is it potentially more important?

We removed the word “more”

9. Fig. 1 has a low resolution. The new caption text seems to belong to Methods.

Fig. 1 is a relatively low resolution video frame because we had to have a large field of view to capture the whole animal (tail included) within the enclosure (the animal did not always hop in the middle of the enclosure). Figure 1 is referred to in the Methods.

10. Fig. 2 caption. Can you also add total trial number?

We added the total number of trials as suggested.

11. I feel the author's response to my previous comments on different speed and non-steady phases trials are informative. It may be useful to mention some of it.

The observations we made concerning non-steady trials were very qualitative and could not be supported quantitatively, so we hesitate to include the observations.

12. The author's response "We have added a reference to support this statement for hopping." Which reference exactly?

Sorry about not being clear about the reference in the response to reviewer's file, which was number (8).

Additional comments by reviewer subsequent to review:

We appreciate the continuing detailed and thorough analysis by the reviewer about the physics of sand intrusion and details about the potential effects contributing to the (ground reaction) force acting on the animal. For this study, we feel including these mechanistic explanations are much beyond our kinematic description of hopping and should be the subject of future studies and papers, like the suggested detailed kinematic analysis of the toe-surface interactions or side wall imaging of the media movement. We only feel comfortable at this point including the excellent suggestion of the reviewer to raise the similarity of the slap phase to landing in hopping. We have put additional text and a reference into the manuscript to state the similarities.

Lines 311-313: "In addition, the landing phase of hopping on sand may be similar to the "slap phase" of locomotion on flowable surfaces, including sand and water, in which the high speed downward impact solidifies the media (6, 19)."

Lastly, we understand that the assumption of a stroke phase is related to many of the concerns of the reviewer that the sand depth was not sufficient. However, the anatomy of the kangaroo rat and hopping motion are quite different than the other animals previously studied. The presence of long toenails and long, coarse hairs between the toes, and unknown flexural rigidity of the toes all create uncertainties about how much rotation of the toes into sand there is.

Appendix B

Dear Associate Editor and reviewers: Please see our responses to the latest review of the manuscript, which follow the reviewers original comments and are italicized.

Associate Editor Comments to Author (Dr Monica Daley):

Associate Editor: 1

Comments to the Author:

Your paper has been reviewed by two experts, who have considered the current manuscript draft as well as the revisions and response to the reviews of the previous Biology Letters version. Reviewer 1 raises fairly minor comments to address; however, Reviewer 2 raises substantial concerns about the potential for boundary effects in the shallow sand substrate, putting the work into the context of relevant previous literature, and providing a more thorough analysis and discussion that fully addresses the valid concerns raised in previous reviews of the work. Considering these comments, I cannot accept the paper in its current form, but I will consider a substantially revised and resubmitted version if the authors can fully address the reviewers' concerns about the analysis and interpretation of the work.

While the authors may not be able to completely address all of the points raised, it should be possible to provide substantive additional analysis, explanation, and discussion in the context of relevant previous literature, keeping in mind that RSOS does not have the same space restrictions as Biology Letters.

*We wish to thank the Associate Editor for the opportunity to address the concerns of the reviewers and patience with regards to the lengthy delay in providing the revision, which was in part due to lab closures and the moving of authors to new institutions. We have made significant efforts and extensive changes to address all reviewers' comments by taking new data characterizing the properties of the sand as well as a more detailed analysis and discussion of the data and study limitations, including relevant previous literature. The Discussion has been completely revised and rewritten in the manuscript. In this revision, we kept in mind that RSOS does not have the have space limitation as Biology Letters and expanded the data and text of the manuscript. All the changes and additions to the manuscript are in *red text*.*

Reviewer comments to Author:

Reviewer: 1

Comments to the Author(s)

The authors have generally responded well to the reviews received when their MS was submitted to Biology Letters and have clarified several matters that were not clearly presented when first submitted. They have also provided a reasonable argument in response to the reviews for the limited kinematics analysis that they perform, in the absence of ground force recordings (which would be quite difficult with their setup), EMGs of muscles and any in vivo muscle forces. They show that basic very general kinematic patterns of the hindlimbs are not significantly different hopping on sand vs a solid substrate. I do think, however, that they need to quantify EL length at mid-support for both condition in the context of arguing that the limb appears more flexed. This may be the case at landing, but the joint angle patterns shown in Fig. 1C do not support this

being the case over much of the support phase. Improved explanation and discussion of this is the only main issue I have regarding the revised MS and the changes made by the authors regarding their findings.

We appreciate the feedback from the reviewer, including the suggestion about the additional analysis of effective limb (EL) length at midsupport. We have added this analysis and put additional text in the Results and Discussion (see reply to l. 174-177 comment below).

Specific comments:

l. 56-58 “In this previous study on lizards, the locomotion was performed in bursts, implying that the movements may not have been during steady locomotion, thus influencing the kinematic patterns.”

Suggest rephrase as “However, locomotion was performed in burst in this previous study of lizards, implying that the movements may not have been during steady locomotion, thus influencing their kinematic patterns.”

We have rephrased the text as the reviewer suggested, but with simpler language.

l. 57 “However, this study examined burst locomotion, ...”

l. 61-61 “Because of the similarity of the task in this previous study to the impact of landing and pushoff during animal hopping”

Suggest rephrase as: “Because of the similarity of this robotic task to the impact of landing and pushoff during animal hopping”

We have rephrased the text as suggested.

l. 63-64: last new sentence - this would seem better made as the topic sentence of the paragraph not as its conclusion.

We would like to keep the first (topic) sentence of this paragraph as originally written because it transitions from the last sentence of the previous paragraph (adaption of motor strategies to terrain substrate) to this paragraph (changes in motor strategies are evidenced in kinematic changes). However, we agree with the reviewer, the sentence appears to be a topic sentence because it starts with “In general”. We replaced “In general” with “However” to signify in contrast with the studies previously mentioned, hopping has not been well-studied. This sentence at the end of the paragraph also serves as a transition to the next paragraph.

l. 70 suggest: “maintain similar general”

We have rephrased the text as suggested.

l. 84 “metatarsophalangeal”

We have rephrased the text as suggested.

l.99-102 “and calibrated for spatial measurements” repeated twice in these sentences

We have corrected the text as suggested.

1.160 “patterns” pl. not singular

We have corrected the text as suggested.

1.166 horizontal and vertical velocities of ?? (hip as proxy for COM, I believe)

We have clarified the text by adding “of the hip”

1.168: ditto - presumably these are for hip as COM estimate

We have clarified the text.

1.173 “gross kinematics” add “that we measured”

We have corrected the text as suggested.

1.174-177: the authors state that the limb appears more flexed hopping on sand, but the joint angles shown in Fig. 1C don’t clearly demonstrate this. To quantify this, they report EL length “at landing”. However, this only reflects how flexed the joints may be at landing; not over the course of stance or mid-support when the greatest GRF would be expected to act. The authors should also test for whether EL length differs at mid support to better evaluate the the limb is more flexed in relation to greater joint loading (rather than just a landing and takeoff). Clearer explanation and rationale for these statements and what is shown in Fig. 1C is therefore needed.

As the reviewer points out, additional support for a more flexed posture while hopping on sand would more clearly demonstrate this point. We tested for a significant difference in EL length at midstance between the sand and solid substrate conditions and the results were ambiguous with $P=0.0625$, which is likely due to the small sample size of $n=6$. We have added this analysis and put additional text in the Results and Discussion.

Results (l. 227-230): “The difference in EL length at midstance was more ambiguous and did not reach the level of statistical significance, with a level of $P = 0.0625$ (sand and solid, 46.1 ± 3.8 mm and 49.6 ± 3.4 mm, respectively).”

Discussion (l. 282-284); “... the limb posture, as assessed by the EL length, was found to be different at landing and during midstance (although not quite reaching statistical significance for midstance).”

1.188 rephrase as: “we presume that hopping on sand imposed”

We revised the text to read:

l. 251 “... we expected that hopping on sand imposed ...”

1.194-195 suggest rephrase as “Thus, it is likely that some neuromuscular compensation occurred to offset...”

We have removed this sentence due to the complete rewrite of the Discussion.

1.196 - break into 2 sentences. "substrate. This hypothesis..."

Also "in vivo force transducers" force of what?? (VAGUE)

We have removed this sentence due to the complete rewrite of the Discussion.

1.208-210 re: centrifugal force being 1/3 body weight - I would suggest this is a good place to acknowledge limitations of 2D kinematics analysis and that 3D kinematics would help to address any ML compensation by the animals.

We have added this limitation to the text. We added:

l. 274-275: "This assumption would have to be confirmed by three dimensional video recordings."

Figures

Sample sizes should be given for the plots in 1C and Fig, 2A,B&C. In Figs 1C & 2C, it would help to identify the stance vs swing phases, even though 'takeoff' is labeled. Also, I don't find the light orange - brownish trace for sand very clear - if made darker or a brighter color it would show up better.

We have modified the figure as suggested by changing to a darker color, adding the sample size, and adding the label "Ground contact" and "Aerial phase" to the figure.

Reviewer: 2

Comments to the Author(s)

This study uses a rotary treadmill and high speed imaging to record and compare the kinematics of kangaroo rats hopping on sand vs. solid surfaces. The authors found that the gross kinematic variables are similar, but a few leg kinematic features are different.

The study is novel in that it is the first to compare bipedal hopping kinematics on sand vs. solid surfaces. However, it suffers several major issues that must be addressed before further consideration.

1. It is not clear if the very shallow sand bed tested really behaves like sand or has substantial boundary effect from the bottom. The kangaroo rat's leg length is on the order of 10 cm (from Fig. 2C), and its foot alone is probably 3 cm long. By contrast, the sand bed is merely 2.5 cm deep. It will not be surprising if the toes can reach the rigid bottom, if the foot undergoes substantial rotation subsurface (see Li, Hsieh, Goldman, 2012, JEB, Fig. 11C). Even if it does not, there can be substantial forces from the solid boundary. This is a well-known effect in confined granular media. Even when an intruding object is still at a substantial distance from the walls that contain granular media, forces already start to go up substantially (e.g., see Li, Zhang, Goldman, 2013, Science, Fig. 2B and Materials & Methods first section). My guess would be that the near solid bottom effect due to the very shallow depth of the sand largely contributes to

the similar overall kinematics observed. Why not try it again with deeper sand (~10 cm or above)? It looks like the treadmill can well accommodate this with only modest modification (taller walls, stronger motor). The observed difference in knee angle etc. are likely evidence that there are still some effect of the sand displacing and dissipating energy, which I predict will be much more obvious if the depth is large enough to remove the solid boundary effect.

The reviewer makes a valid and important criticism that the depth of the sand may not be sufficient to prevent an interaction with the underlying solid surface. To determine whether this may have been the case, we did additional experiments to characterize how the sand properties changed with depth, using a protocol similar to Li et al., 2012 JEB. The methods and results have been added to manuscript along with an additional figure as described below. Our finding with the sand from the Mojave desert and using a penetrator with the same dimensions of a kangaroo rat foot is that properties of the sand do not change statistically after the sand depth has reached 1.5 cm. Further, our results support the idea that the combination of kangaroo rats relatively large toe area and the properties of the Mojave sand is the reason that there was no change in kinematics (see reply to comments below). In addition, the reviewer suggests, rotation of the foot (like the zebra-tailed lizard) could allow the solid underlying surface to influence the results. The stiffness measurements of the sand indicate that for 3 N force (approximately equal to the ground reaction during hopping), there was less than 0.7 mm of penetration (see Fig. 4), and this supports our observations that the foot did not come close to bottoming out (see additions in Results).

Methods:

“To verify that the depth of sand on the treadmill was sufficient to prevent boundary effects of the solid surface below, we made sand penetration measurements for different sand depths, similar to the testing protocol of Li et al. (2). Specifically, a custom-made penetrator was mounted on servo system (model 309C; Aurora Scientific; see technical note 300C-I-T01 for details on the setup). The penetrator was made out of wood and had the rectangular dimensions of 2.5 x 1.25 cm, which was based on dimensions of an ink print of the toe segment of an average-sized kangaroo rat. During steady speed hopping, only the toes contact the surface. To replicate the landing of the toes on the sand during hopping, we set the servo to a constant velocity of 50 cm/s, which was the average vertical velocity of the foot at landing. The penetration force was limited to 3.0 N, which was based on the maximum ground reaction force data per leg observed during kangaroo rat hopping (12). Penetration tests were repeated five times for sand depths of 0, 0.5, 1.5, 2.0, 2.5, 3.0, 3.5, 4.0, and 5.0 cm. To increase the consistency of the measurements, the penetrator was positioned above the sand surface and slowly lowered until a very small pre-load (0.2 N) was reached. This procedure and amount of pre-load made the surface more uniformly flat because the small pre-load smoothed out any irregularities while not compressing the sand. The sand was aerated after each trial by gentle raking and then re-leveled.

To quantitatively characterize the sand, a measurement of the penetration resistance was made for each trial of data. Because the plot of force versus length was roughly linear until the force was 2.5 N in all trials (see Results), penetration resistance was estimated as:

$$PR = \frac{(2.5 - F(t = 0))}{(X(F = 2.5) - X(t = 0))} \quad (1)$$

where PR is the penetration resistance (N/mm), F is the measured force (N), X is the position of the penetrator (mm), and t is time. Note that the numerator of eqn. (1) is the difference of 2.5 N and the initial force (about 0.2 N) and the denominator is the difference between the position when the $F = 2.5$ N and the initial position of the servo. We determined whether there was a significant difference in the PR for the sand depths by using an ANOVA followed by multiple pairwise comparisons (Tukey's honest significant difference criterion; "multcompare" in the MATLAB statistics toolbox (MATLAB 2019b; Mathworks)). Significance level was taken to be 0.05."

Results:

"The sand penetration produced a force-position curve from which the PR was estimated (Figure 4A). The PR was highest for the wooden surface (0.0 mm of sand) and decreased as the sand depth increased until a depth of 1.5 mm. Increasing the depth beyond 1.5 mm did not change the PR (Fig. 4B). The ratio of the PR of the wood surface to PR of the sand depth used in the experiments (2.5 cm) was 2.8:1. The ANOVA with sand depth as the factor was statistically significant ($P < 0.00001$) and the pairwise comparisons showed that PR s for sand depths of 0.0 and 0.5 were statistically different than the PR for any other depth, and the PR s for any sand depths greater than or equal to 1.5 mm were not statistically different. These results support that the sand depth of 2.5 cm used was sufficient to avoid any boundary effects of the wooden surface below the sand."

Figure 4 is a new figure showing the data from testing the sand.

2. The author neglects very relevant previous work, especially those from Dan Goldman's lab (e.g., Li, Hsieh, Goldman, 2012, JEB, Qian et al, 2015, B&B) as well as others (e.g., Zhang et al, 2018, PLOS One). Even though the study here is unique in that it concerns bipedal hopping, the main ideas presented in this manuscript (e.g., similar gross kinematics and center of mass oscillation patterns, energy dissipation, compensation by extending legs more to do more muscle work) have been well worked out in these previous studies through more thorough experiments and modeling. This is especially concerning because running as studied in these previous works have observed essentially the same SLIP-like CoM dynamics which hopping (at least on rigid ground) also follows. As good scholars (specifically here but also in general), the authors need to acknowledge previous contributions and strive to carry out their work (both experimentally and during analysis and modeling) to demonstrate how it provides new insight in the context, rather than ignore some of the most relevant literature in order to make it appear like the work is brand new in arriving at these ideas.

We agree with the reviewer that we should include more context for our study by discussing previous relevant studies, both experimental and modeling in nature. Note that our manuscript was originally submitted to Biology Letters, which has a very limited length (2500 words (including references) and 2 figures), which constrained us to only a few references. Because we are no longer under this limitation (see Associate Editor's note), we are able to include the relevant studies as suggested by the reviewer. We also wish to thank the reviewer for pointing out the Qian et al. study, which provided valuable information about how we could quantitatively analyze the effect of the substrate given the ground contact area and weight of the animal.

In the Introduction, we added:

“In running zebra-tailed lizards, it was estimated that the hindlimb muscles performed three times more work on a granular substrate than on a solid surface (2).”

“In another study of lizard running, stride length decreased by 15% on the granular versus solid substrate, and the limb posture shifted from a digitigrade to a plantigrade position on solid versus granular substrate, respectively (2, 4). In ostriches, the kinematics of the toe joints were different while running on sand versus solid substrates, with more range of motion greater on the sand surface (5).”

“... or, the combination of contact surface area (i.e., toeprint area), sand properties, and animal weight were such that animals did not need to compensate.”

In the Discussion, we added:

“First, a previous study of zebra-tailed lizards estimated that muscles do three times the amount of work while the animals run on a fluidized bed of granular material versus on a solid surface.”

“Although bipedal hopping differs substantially for other modes of locomotion studied, these two reasons suggest that the animal would be doing work on the sand and as a result, we predicted that the kinematics of the locomotion pattern would change with the change in substrate, similar to other animals (2, 4, 11, 14).”

“In an insect study, decreases in the vertical position of the body COM occurred while running on more compliant surfaces, similar to the more crouched posture observed in this study (17).”

“The second potential mechanism for the lack of changes in hopping kinematics is that neuromuscular adaptation was not needed due to the physical characteristics of the sand, the area of toeprint during the contact phase, and weight of the kangaroo rats. With regards to the sand properties, variations in sand composition, granule size, and granule shape all influence different mechanical characteristics (18). Our measurements of the sand from the kangaroo rats’ habitat included angle of stability, bulk density, and penetration resistance (PR). Interestingly, in a survey of different sand with a specific ranges of particle sizes, the angle of stability (33° and bulk density (1.56 gm/cm³) of the sand used in our study were very similar to the sand (34° and 1.532 gm/cm³, respectively) on which steppe-runner lizards had the highest locomotor performance (maximum velocity and acceleration) (11). Thus, it is possible that the properties of the Mojave sand are well suited as a substrate for high performance locomotion to occur on.”

A detailed study of the interdependence of sand properties, contact area of the foot, and animal weight and limb length for locomotor performance was conducted by Qian et al. (14) using a fluidized bed of granular material. This study utilized a hexapod robot, three species of lizard, geckos, and ghost crabs to measure the locomotor performance for a diverse range of anatomical morphologies and sizes on a wide range of substrates, with the objective of generalizing how substrate properties, morphology, and size interact to affect locomotor performance. A unitless parameter, called the leg penetration ratio, \tilde{d} , was formulated as:

$$\tilde{d} = F / (k * l) \quad (2)$$

where F is the force exerted on contact area, k is the penetration resistance (PR), and l is the length of the limb. For applying equation (2) in the Qian et al. study, F was taken as the weight of the animal, k is an experimentally based parameter that depends upon the substrate and scales linearly with contact area (4), and l was taken as hip to ankle length. The importance of \tilde{d} is that for $\tilde{d} < 1$ (called the “insensitive region”), average locomotion speed was unaffected by the

substrate. Conversely, for $\tilde{d} > 1$, increasing values of \tilde{d} resulted in a steep decrease in average locomotion speed.

For kangaroo rats hopping on Mojave sand, we used a value of $F = 3.0$ N because that was the maximal ground reaction force observed (12), $k = 3.6$ N/mm was the value found for the sand depth of 2.5 cm (Figure 4B), and $l = 50$ mm was approximately the minimum EL length recorded (Figure 3), resulting in a value of $\tilde{d} = 0.0056$. Note that although we used values of F and l that maximized the value of \tilde{d} , the value of \tilde{d} was still substantially less than the critical value of 1.

The calculated value of \tilde{d} provides strong support that the properties of the Mojave sand, toeprint contact area, and weight of the animal (as reflected in the value of F) combine to make kangaroo rat hopping performance insensitive to the sand substrate. Specifically, the size of toeprint area plays an important role because larger area increases the PR proportionally. In desert kangaroo rats, the toe segment dimensions were measured to be 2.5 x 1.25 cm, approximately 16% of their total leg length (summed segment lengths) (19), and all kangaroo rat species have a greatly elongated foot compared to similarly size quadrupedal mammals (13). Potentially more important is that the toes of desert kangaroo rats are covered in dense fur, filling the space between the toes and increasing the total surface area. More generally, this species is highly specialized because it is only found in habitat with soft sand dunes (20). This study confirms that desert kangaroo rats have morphological adaptations that enable them to maneuver through soft sand with high locomotor performance.”

Considering this, I strongly encourage the authors to analyze their entire dataset at five speeds to see if some real novel observations and insights remain to be discovered. Another suggestion is evaluating how kinematics changes with speed during the non-steady phases of the trials on sand. The authors mentioned that these recorded trials are not always in steady state due the sand being accelerated. It would be interesting to test if the animal’s kinematics shift during this transient and infer what they may be doing in terms of muscular control and force generation, in the context of previous literature.

The reviewer suggests that it may be possible to analyze the other speeds used. Indeed, our original experimental design was intended to examine the effect of speed, which is the reason that we tried five speeds. Unfortunately, the animals preferred a certain speed (the speed for which the results were reported) and at other speeds, they often struggled to hop regularly (we have been able to get animals to hop at other speeds on a regular belted treadmill but have not been as successful on this treadmill). Thus, we had a relatively small number of “valid” hops (see manuscript for definition of valid hop) at those other speeds (at some speeds there were no valid hops for a specific animal) which makes any statistical analyses impracticable. The reviewer also suggests that it may be possible analyze the non-steady phase of hopping, during the acceleration phase. Because we are using wild animals, there was variation of how each animal reacted to the start of the treadmill. In examining the videos, some animals dug in their heels (literally) before hopping and others hopped immediately (about half and half). Thus, there is no common mode of acceleration among all animals and it would not have been practical to perform statistical analyses. In summary, although analysis of different speeds and non-steady hops might have revealed how the animals were generating hopping under the different conditions, the current data set does not have enough consistency and/or samples to make statistical evaluations possible.

3. There are a number of vague statements and assumptions that need to be better explained/justified, without which the claims are undermined. To name a few:

a. The authors repeatedly state they chose speeds that yield the “most consistent hopping”. Please quantify what you mean.

We agree that “consistent” should be better defined. We looked for valid hops in the video recordings and the speed that yielded the most valid hops was the one that we analyzed. This has been clarified in the text at line 113.

b. Line 131: some hops—how many? Is it the majority of the hops of these trials? If so then this contradicts the argument that the treadmill address the non-steady locomotion previous work suffers from.

We agree that “some” should be better defined. We included the percentage of hops where the hop where the treadmill was not yet at steady-state speed.

In the Methods (l. 140): “(about 12% for the sand trials).”

c. Line 133: have you looked at the videos carefully (even try manual tracking) to verify the assumption of foot speed being equal to VTRT speed is valid? Previous works (e.g. ones from Goldman’s group) have found substantial slippage of foot during running on sand.

d. Line 137: Can you show the data to show that they agree?

We manually digitized the videos and as best we could see, there is minimal slipping in the trials that we included in our analysis. However, we also recognize that because the foot was partially obscured by the sand, thus we cannot be entirely sure of the amount of slippage. We have added text that states that we assumed that there was minimal slippage.

In Methods (l. 143-145): “This assumption presumes that there was minimal slippage of the foot on both surfaces, but this could not be verified on sand due to the sand obscuring visualization of the entire foot.”

4. The authors did not seem to have taken reviewers’ valid concerns as seriously as they should.

If it seemed like we did not take the reviewers’ comments seriously, we apologize for the ambiguity and lack of completeness. We tried to address each comment completely, but for some concerns, we did not have the data to adequately address those concerns.

a. Throughout the manuscript and response letter, there are many places where the authors refer to data not shown or “unpublished data” to justify their statements. They should add supplementary material or at least include them in response to reviews.

In reviewing the previous Reply to Reviewers, we used unpublished data to justify two aspects of our methodology. First, we compared joint angle estimates using the methodology and stated to it compares to estimates from unpublished x-ray cinematography. We should have instead stated that the methodology is a method that has been used in several published studies and accepted as the standard methodology. We have included two references in the text to clarify this point (line 128). Second, we stated that the preferred speed on the treadmill was consistent with the speed from over ground trials. We added the reference for this statement (l. 201-202).

b. There are also many places where the authors simply address concerns by stating “we prefer not to” without giving convincing arguments.

In the previous Reply to Reviewers, we once used the phrase “we prefer not to [discuss other substrates in order to not confuse the reader]” in response to the reviewer’s mention that hopping on sand might be analogous to basilisk lizards running on water. Our intent was not to simply dismiss this suggestion but given the differences between lizard running and kangaroo rat hopping and between the properties of sand and water, we believed that the comparison would have been too speculative.

c. Just because a journal imposes word limit does not mean one should not address critical concerns. There is quite a bit of repetition of ideas throughout (treadmill limitation, mechanical constraint by sand, to name a few). This is substantial considering how short the manuscript already is. The authors should remove repetition and instead address concerns.

Because there is no longer the constraining limitation of the Biology Letters format, we can address more of the concerns in detail. The reviewer may be referring to referee 1’s comment about potential differences in kinetics and dynamics, to which we replied, “these more involved analyses were beyond the scope of our study and would be problematic to present within the word limitation of the journal.” We did not mean to use the word limitation as an excuse. Our main point was that kinetic and dynamic analyses were beyond the scope of our experimental design and objectives, namely, without any force plate data, the analyses were not possible.

As for repetition, we wanted to convey as much information about our methods so that the reader was cognizant of the studies limitations, especially with the treadmill because it is a novel device. We would welcome any specific suggestions about how to remove any repetition.

Minor:

I do not agree with the last sentence of abstract. Even without considering the potential flaw in using sand too shallow, the mere kinematic resemblance of a crouched posture does not suggest similar mechanical adaptations. On sand the crouched posture is more of an adaptation for legs to extend and push harder to do work to compensate for the energy loss into sand (which will likely be more obvious if the authors used deeper sand). By contrast, on rough terrain, I suspect it would be more of an adaptation to maintain dynamic stability.

Our intent with this last sentence was to point out that there is a similarity of a more crouched posture for the two cases. We have taken this comparison out of the abstract, changing the last sentence to focus on our new analyses based upon the measurements on the sand:

l. 32-34: “Measurements of the penetration resistance of the sand showed that the combination of the sand properties, toeprint area, and kangaroo rat weight was likely the reason for the similarity.”

Line 46: Incorrect. Even on rigid ground, legs do lose energy when colliding with the surface. *We agree that this statement was made in error. We changed the text to read:*

“... rigid surface that animals lose minimal energy to during impact or during pushoff ...”

“Angle of stability” should be “angle of repose”. Also, the authors do not seem to understand this well. A higher angle of repose results from particles having larger particle-particle friction, which means that the bulk of granular media is stronger in withstanding external load. So it is only when a granular media is oriented at a slope close to the angle of repose that it becomes very weak and avalanches easily. In Line

We did not adequately explain the measurement that we performed. We put the sand in an aluminum tray and tilted the tray until the sand started to flow. We believe that this measurement is for the angle of stability, as defined in Albert et al., Physical Review E, 1997, which states, “a granular material is stable when its container is tilted slightly as long as the top surface is at a slope less than the angle of maximal stability θ_m .” We have included these details as well as the reference. Also, we choose to report angle of stability rather than angle of repose so that we could compare the measurements of our substrate to the measurements reported for other substrates in Bergmann et al. (2017),

In Methods:

l. 164-168: “To characterize the properties of the sand used in the experiments, we first measured the angle of stability by placing and leveling sand horizontally on an aluminum tray and tilting the tray until the sand began to flow (10, 11). The angle of stability was found to be 33°. Next, we estimated the bulk density of the sand by weighing a 50 ml graduated cylinder filled with sand (11). The bulk density was calculated to be 1.56 gm/cm³.”

Line 99: What do you mean by “calibrated for spatial measurements”? Only one camera was used so there is no 3-D calibration. What is the lens focal length used?

The recorded video was calibrated to convert pixel to cm. We have added this to the text. We also added the focal length of the camera used as suggested.

What is the stride frequency of the animal? Is a 25 Hz cut off frequency sufficient for filtering? *The average hop frequency was 6 Hz (the inverse of the hop period). The 25 Hz cut off frequency was selected because it removed the high frequency noise associated with the digitization while keeping as much of the low frequency components of the movements as possible, similar to other studies in our laboratories. We included this information in the text.*

l. 133-134: “(cutoff of 25 Hz which was sufficient for the 6 Hz hopping frequency (see Results))”

What are the statistical tests being used exactly?

The test that we used was the non-parametric two-tailed Wilcoxon signed-rank test, which is stated in the Methods section (lines 158-161).

Line 163: The authors assumed that hopping is SLIP-like here. This is not necessarily true. It leaves the impression that the authors were looking for certain features in the data to “prove” something they assumed. What if the hip is not close to the CoM? Also, I don’t quite understand how the hip joint angles alone show the SLIP-like dynamics. Please elaborate.

We stated that we expected that “the horizontal velocity of the COM should decelerate upon landing, accelerate toward takeoff, and be near constant in the aerial phase” which is not model dependent but a consequence of the ground reaction force being horizontally backwards (relative to the movement of the animal) at landing, horizontally forwards at takeoff, and zero in the aerial phase, respectively. We have added a reference to support this statement for hopping. With regards to the using the hip as an approximation of the COM, we tried to be clear that it is the best approximation of all our markers. However, as the reviewer points out, the hip marker location (not joint angle) is likely different than that of the COM. The actual COM lies approximately 2-3 cm anterior to the hip, and based on the animals crouched posture, the COM does not likely shift substantially, relative to the hip, during hopping. Therefore, the hip velocities and accelerations are likely to be close to that of the COM velocity and acceleration because the hip marker and COM are on the torso of the animal’s body and are a constant distant from each other. We have added a statement in the text stating this assumption.

In Results:

l. 213-216: “Although the actual COM lies approximately 2-3 cm anterior to the hip in animals with a crouched posture, the COM position does not likely shift substantially relative to the hip position during hopping, thus we assumed that the velocities and accelerations should be similar for the hip and COM.”

Line 167-169: Can you show these data?

We added Table 1 to show these data.

Fig. 1. It’d be useful to add definition of coordinate system and define EL, length, y length, and the angles being plotted in Figs. 1 and 2. Why not plot a line and band (s.d.) to show take off moment data?

We have enlarged the figure in Figure 1B to more clearly show the variables of interest. As for including a band showing the s.d. of the take off time, because there are two sets of data shown (sand and solid), and when we put the s.d. bands on the graph, they overlaid so much that the s.d. bands were not clearly distinguishable. Instead, we showed the mean and s.d. for the sand and solid data by a line (mean) with an error bar (s.d.) for each data set.

Fig. 2. Need to explain error bars are s.d.

We add the explanation as suggested.

I believe by “cumulative errors”, Reviewer 1 means errors accumulated spatially, not temporally. This still needs to be addressed.

The reviewer is correct that we did not address the spatial error. It is true that there is error in our estimation of the position of the hip joint and ankle joint (true for all marker based position tracking) but that error is random and relatively small due to the skin being tight. Therefore, there would not be any systematic error that accumulates over time.

We added to the Methods:

l. 130-132: "Skin markers over the ankle and hip joints had minimal random motion artifact because the skin is relatively tight over those joints, thus any error in the knee angle estimate should not accumulate spatially."